# Insights into the missing apiosylation step in flavonoid apiosides biosynthesis of Leguminosae plants

Hao-Tian Wang[1,4], Zi-Long Wang [1,4], Kuan Chen[1], Ming-Ju Yao[1], Meng Zhang[1], Rong-Shen Wang[1], Jia-He Zhang[1], Hans Ågren[2], Fu-Dong Li[3], Junhao Li[2] ✉, Xue Qiao [1] ✉ & Min Ye [1] ✉

Apiose is a natural pentose containing an unusual branched-chain structure. Apiosides are bioactive natural products widely present in the plant kingdom. However, little is known on the key apiosylation reaction in the biosynthetic pathways of apiosides. In this work, we discover an apiosyltransferase GuApiGT from *Glycyrrhiza uralensis*. GuApiGT could efficiently catalyze 2″-*O*-apiosylation of flavonoid glycosides, and exhibits strict selectivity towards UDP-apiose. We further solve the crystal structure of GuApiGT, determine a key sugar-binding motif (RLGSDH) through structural analysis and theoretical calculations, and obtain mutants with altered sugar selectivity through protein engineering. Moreover, we discover 121 candidate apiosyltransferase genes from Leguminosae plants, and identify the functions of 4 enzymes. Finally, we introduce *GuApiGT* and its upstream genes into *Nicotiana benthamiana*, and complete de novo biosynthesis of a series of flavonoid apiosides. This work reports an efficient phenolic apiosyltransferase, and reveals mechanisms for its sugar donor selectivity.

The naturally occurring D-apiose is a unique branched-chain pentose with a tertiary alcohol group, and is considered as "one of nature's witty games"[1]. The name "apiose" is derived from apiin (apigenin 7-*O*-apiosyl(1→2)-glucoside), the first apiose-containing natural product isolated from parsley in 1843[2]. In plants, apiose is synthesized as uridine diphosphate-apiose (UDP-Api) from UDP-glucuronic acid (UDP-GlcA) catalyzed by UDP-apiose/UDP-xylose synthase (UAXS)[3,4]. It is also a key component of complex cell wall polysaccharides, which play important roles in plant growth and development[5]. The apiose-containing plant pectic polysaccharide RG-II has been a component of human diet for a long history, and exhibits notable benefits to human health[6].

More importantly, apiose is an important building block of various natural products. Around 1200 apiosides have been identified from plants[1], thus far, including phenolic glycosides (e.g. flavonoids, coumarins, and lignans), triterpenoid saponins, and cyanogenic glycosides. Among them, flavonoid apiosides represent the largest group, and are believed to play hormone-like roles in plant growth regulation[7]. In the structures of flavonoid apiosides, the apiosyl residue is usually linked to 2″-OH of sugar moieties through a β-*O*-glycosidic bond. It may also be linked to 3″-OH or 6″-OH of sugar moieties or the flavonoid skeleton directly[8]. Leguminosae is one of the most frequently reported plant families with flavonoid apiosides[1]. *Glycyrrhiza uralensis* Fisch. is a worldwide popular medicinal plant of the Leguminosae family. Its roots and rhizomes are used as the famous Chinese herbal medicine Gan-Cao (licorice)[9]. Licorice contains abundant flavonoid apiosides (around 1% of the dry weight) as bioactive compounds, particularly liquiritin apioside and isoliquiritin apioside

[1]State Key Laboratory of Natural and Biomimetic Drugs, School of Pharmaceutical Sciences, Peking University, 38 Xueyuan Road, Beijing 100191, China. [2]Department of Physics and Astronomy, Uppsala University, SE-751 20 Uppsala, Sweden. [3]National Science Center for Physical Sciences at Microscale Division of Molecular & Cell Biophysics and School of Life Sciences, University of Science and Technology of China, Hefei 230026, China. [4]These authors contributed equally: Hao-Tian Wang, Zi-Long Wang. ✉e-mail: junhao.li@physics.uu.se; qiaoxue@bjmu.edu.cn; yemin@bjmu.edu.cn

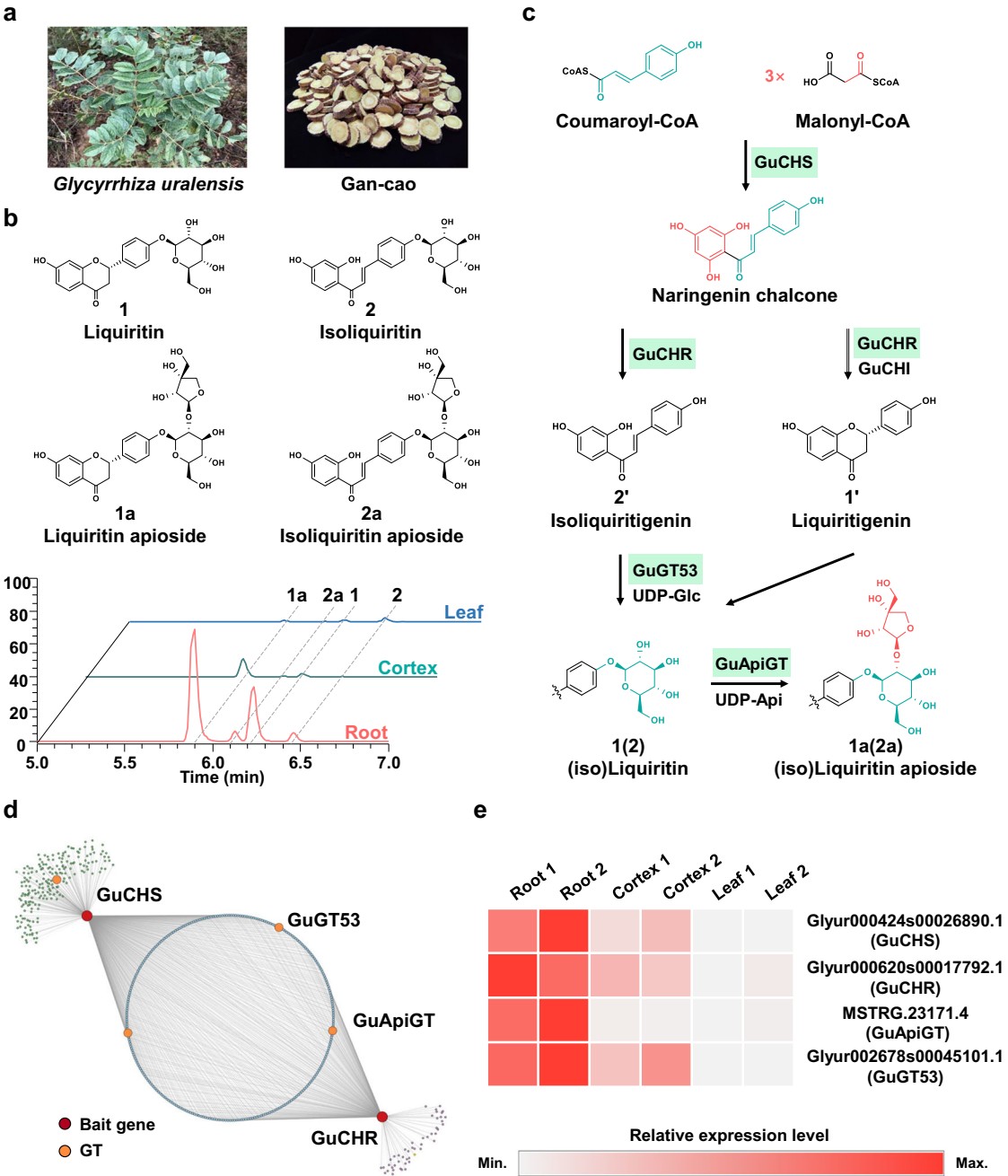

**Fig. 1 | Bioinformatic analysis of candidate apiosyltransferase genes from *Glycyrrhiza uralensis*. a** Pictures of *Glycyrrhiza uralensis* plant and the Chinese herbal medicine Gan-Cao. **b** LC/MS analysis of the roots, cortexes, and leaves of *G. uralensis*, showing extracted ion chromatograms (XICs) of **1**, **2**, **1a** and **2a**. **c** The proposed biosynthetic pathway of **1a** and **2a**. CHS, chalcone synthase; CHR, chalcone reductase; CHI, chalcone isomerase. **d** Co-expression analysis of *G. uralensis* transcriptomes using *GuCHS* and *GuCHR* as bait genes. The circle of dots represents genes co-expressed with *GuCHS* and *GuCHR*. **e** Expression levels of candidate genes in the transcriptomes of different plant parts of *G. uralensis*. Two replicates were tested for each part (*n* = 2).

(Fig. 1a)[10]. Among them, liquiritin apioside (liquiritigenin 4′-*O*-apiosyl(1→2)-glucoside) shows potent antitussive activities[11].

Currently, flavonoid apiosides are mainly obtained through extraction and purification from plants. The procedure is laborious and time consuming. The unique structure of apiose has attracted organic chemists to develop new methods to synthesize apiosides. However, these methods usually take multiple steps, and need expensive metal catalysts[12]. In plant biosynthesis, the formation of glycosidic bond is usually catalyzed by uridine diphosphate-dependent glycosyltransferases (UGTs)[13]. The UGT-mediated glycosylation reactions take only one step, and show high catalytic efficiency and selectivity. Thus far, a big family of plant UGTs have been reported[14], and most of them accept popular sugar donors such as UDP-glucose (UDP-Glc), UDP-xylose (UDP-Xyl), UDP-galactose (UDP-Gal), UDP-rhamnose (UDP-Rha), UDP-arabinose (UDP-Ara), and UDP-glucuronic acid (UDP-GlcA)[14,15]. It is noteworthy that no UGTs could accept UDP-Api as sugar donor except for the recently reported UGT73CY2 which could accept a triterpenoid saponin substrate[16].

In this work, we report an efficient phenolic apiosyltransferase GuApiGT from *G. uralensis*, and dissect mechanisms for its sugar donor selectivity towards UDP-Api through crystal structure analysis, theoretical calculations, and mutagenesis. A key motif (RLGSDH) of

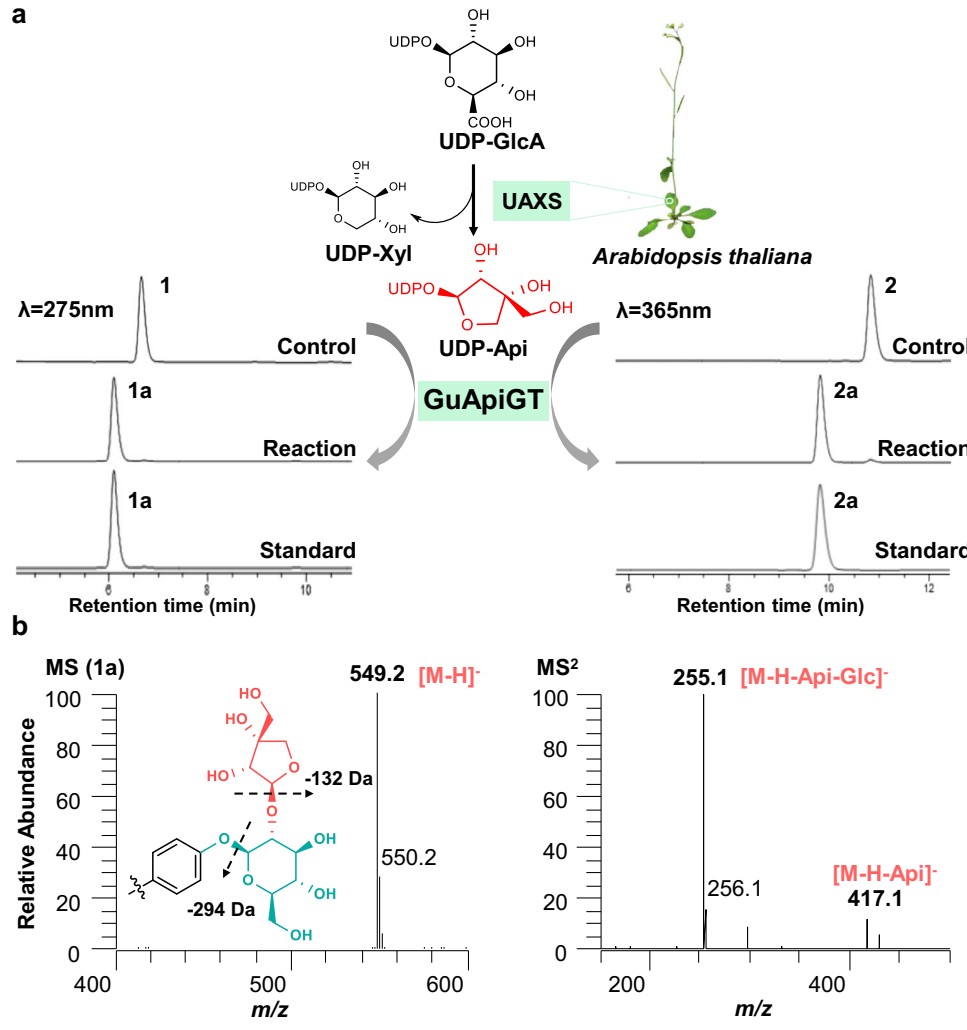

**Fig. 2 | Functional characterization of GuApiGT. a** Catalytic function of GuApiGT using **1** and **2** as sugar acceptors. The acceptors were incubated with GuApiGT and an UDP-Api supply system (UDP-GlcA, UAXS, NAD⁺) in pH 8.0 (50 mM NaH₂PO₄- Na₂HPO₄ buffer) at 37 °C for 3 h. **b** (-)-ESI-MS and MS² spectra of **1a**. For compounds identification, see Fig. 1.

GuApiGT led to the discovery of a group of apiosyltransferases from Leguminosae plants. Furthermore, we realized the de novo biosynthesis of a series of flavonoid apiosides in *Nicotiana benthamiana*.

## Results and discussion

### Bioinformatic analysis

*G. uralensis* contains (iso)liquiritin apioside as major compounds. The high yield strongly suggests the presence of apiosyltransferases in this plant. To discover the apiosyltransferase gene, we conducted co-expression analysis[17,18]. As the contents of (iso)liquiritin apioside (**1a** and **2a**) in the roots were higher than those in the cortex and leaves, transcriptomes of these three parts were analyzed (*n* = 2, Fig. 1b). All high expression genes (FPKM ≥ 20) in the roots were used as candidates. *GuCHS* and *GuCHR* were used as the 'bait', because they are key genes involved in the biosynthesis of (iso)liquiritigenin (**1′** and **2′**), which are precursors of (iso)liquiritin apioside (Fig. 1c).

Through co-expression analysis, a total of 289 genes were obtained (r ≥ 0.8, Spearman correlation coefficient) (Fig. 1d). Pfam, NR, and Swissprot databases annotated four candidate UGT genes. Aside from two previously reported triterpenoid glycosyltransferases (GuRhaGT and UGT73P12)[19,20], the other two unknown genes showed very similar expression patterns with *GuCHS* and *GuCHR* (Fig. 1e). In the phylogenetic tree, MSTRG.23171.4 was clustered with flavonoid 2″-*O*-

glycosyltransferases including ZjOGT38[21] and TcOGT4[22], and was considered as the candidate apiosyltransferase gene (Supplementary Fig. 1).

### Molecular cloning and functional characterization of GuApiGT

Based on the above bioinformatic analysis, we cloned MSTRG.23171.4 from the cDNA of *G. uralensis* by RT-PCR (Supplementary Data 1). The gene in pET28a(+) vector was then expressed in *E. coli* BL21(DE3) and purified by His-tag affinity chromatography (Supplementary Fig. 2). As uridine diphosphate-apiose (UDP-Api) is commercially unavailable, we introduced the UDP-apiose/UDP-xylose synthase (UAXS) of *Arabidopsis thaliana* into the enzyme catalysis system (Fig. 2a)[3]. This system was used to provide UDP-Api in follow-up experiments. Liquid chromatography coupled with mass spectrometry (LC/MS) analysis and reference standards comparison indicated MSTRG.23171.4 almost completely converted liquiritin (**1**) and iso-liquiritin (**2**) into their 2″-*O*-apiosides **1a** and **2a**, respectively (Fig. 2b). Although UAXS could also produce UDP-Xyl, no products were observed when UDP-Xyl was added (Supplementary Fig. 3). These results confirmed MSTRG.23171.4 as an apiosyltransferase, and it was named GuApiGT. To our best knowledge, this is a previously unidentified apiosylation pathway for the biosynthesis of phenolic apiosides.

The gene sequence of GuApiGT (GenBank accession number OQ201607) contains an open reading frame (ORF) of 1365 bp encoding 454 amino acids (Supplementary Table 1). It is named as UGT79B74 by UGT Nomenclature Committee. The biochemical characteristics of recombinant GuApiGT were investigated using **2** as the acceptor. GuApiGT showed its maximum activity at pH 8.0 and 37 °C. Some divalent cations could suppress the catalytic activities (Supplementary Fig. 4). Kinetic analysis demonstrated the $K_m$ value of $2.59 \pm 0.23$ μmol·L$^{-1}$ for **2**, at the presence of saturated UDP-Api. The $k_{cat}$ value was 0.11 s$^{-1}$, and the $k_{cat}/K_m$ was 0.042 s$^{-1}$·μmol$^{-1}$·L (Supplementary Fig. 5).

To explore the catalytic promiscuity of GuApiGT, 65 substrates (**1**-**65**) were tested. LC/MS analysis revealed that GuApiGT showed high substrate promiscuity and high catalytic efficiency. It could accept 37 glycosides (**1**-**37**) of flavonoids, lignans, or coumarins, but not free flavonoids (**52**-**57**, Fig. 3, Supplementary Fig. 6, and Supplementary Tables 2, 3). The products were identified as *O*-apiosides according to the diagnostic fragment ions [M-H-132]$^-$ and [M-H-132-162]$^-$ in the MS/MS spectra (Supplementary Figs. 7–42). For 14 substrates, the conversion rates were >80%.

GuApiGT mainly catalyzed the apiosylation of flavonoid 7- or 4′-*O*-glycosides. The aglycones could be flavanones (**1**, **3**-**6**), chalcones (**2**, **7**-**8**), flavones (**9**-**22**), isoflavones (**23**-**27**), flavonols (**28**-**31**), and dihydrochalcone (**32**). For 7,4′-di-*O*-glycosides like **8**, **19** and **29**, two products were observed, indicating apiosylation at either site. GuApiGT could also catalyze flavonoid 5-*O*-glycoside (**20**), but not 3-*O*-glycosides (**38**-**46**, Supplementary Fig. 6). It is noteworthy that GuApiGT could accept flavone 6-*C*-glycosides (**21** and **22**) and xanthone *C*-glycosides (**33** and **34**), but not isoflavone 8-*C*-glycoside (**47**) or flavone di-*C*-glycosides (**48**-**51**). It could not recognize other types of glycosides including triterpenoid glycosides (**58**-**65**).

To fully identify structures of the products, we purified six 2″-*O*-apiosides (**6a**, **15a**, **24a**, **27a**, **32a**, and **35a**) from scaled-up catalytic reactions. All the products are unreported compounds except for **15a**[23]. Their structures were established by HR-ESI-MS, together with 1D and 2D NMR spectroscopic analyses (Supplementary Figs. 43–81). The $^{13}$C NMR and DEPT spectra showed additional signals at $\delta_C$ 108.8 (C-1‴, CH), 76.2 (C-2‴, CH), 79.4 (C-3‴, C), 74.0 (C-4‴, CH$_2$), and 64.2 (C-5‴, CH$_2$), which are characteristic for an apiosyl group. In the HMBC spectra, the long-range correlation between H-1‴ and C-2″ indicated the apiosyl moiety was attached to 2″-hydroxy of the glucose residue.

**The RLGSDH motif is critical for the selectivity towards UDP-Api**
Interestingly, GuApiGT showed strict sugar donor selectivity towards UDP-Api. It could not recognize seven other donors (Fig. 3c). In order to dissect the mechanisms, we obtained the apo crystal structure of GuApiGT with a resolution of 2.2 Å (Fig. 4a and Supplementary Table 4). Due to the low amino acid sequence identity with reported structures, the structure of GuApiGT was solved by molecular replacement with the help of AlphaFold2 simulation (Supplementary Fig. 82)[24]. The crystal contains two highly similar molecules with a root mean square deviation (RMSD) of 1.1 Å, and adopts a canonical GT-B fold consisting of two Rossmann-like β/α/β domains that face each other and are separated by a deep cleft. The N-terminal domain (NTD, residues 1-242 and 436-454) and the C-terminal domain (CTD, residues 243-435) are responsible primarily for sugar acceptor and sugar donor binding, respectively[25].

It was regretful that we failed to obtain complex structures after many attempts including soaking experiments. Fortunately, the location of UDP in reported UGT complex structures is highly conservative (Fig. 4b and Supplementary Table 5). Based on the structures of GgCGT/UDP-Glc, GgCGT/UDP-Gal, and UGT89C1/UDP-Rha, we simulated the UDP-Api binding pocket of GuApiGT (Fig. 4c)[26,27]. It is noteworthy that a part of UDP-sugar binding region of GuApiGT is different from that of GgCGT (glucosyltransferase), SbCGTb

(arabinosyltransferase)[28], or UGT89C1 (rhamnosyltransferase)[27]. This region is composed of the R$^{368}$L$^{369}$G$^{370}$S$^{371}$ loop and the start of α helix (D$^{372}$H$^{373}$), which forms a large secondary structure compared with other UGTs (Fig. 4d)[26–30]. Moreover, the plant secondary product glycosyltransferase box (PSPG box) of GuApiGT contains 45 amino acids due to the additional S371 residue (Supplementary Fig. 83). In contrast, this highly conserved box for all previously reported plant UGTs contains 44 amino acids[14,25].

To analyze the potential interactions of UDP-Api with key residues, we obtained the initial GuApiGT/UDP-Api complex structure by superimposing the UDP part of UDP-Api to the reported binding pocket of other UGTs. Subsequently, the sugar acceptor was docked into the active site using the Glide module in Schrodinger Suite (Supplementary Fig. 84). Constraints were added to make the acceptor's glucose moiety oriented to UDP-sugar moiety. To optimize the configuration of ligands in GuApiGT, we conducted 100-ns MD simulations (Supplementary Fig. 85)[31]. Representative snapshots of GuApiGT/UDP-Api/**2** complex model indicate that D372, H373, and I136 could form hydrogen bonds with the apiose OH group (Fig. 4e and Supplementary Fig. 86). Moreover, R368 could change its initial state, and the side chain could flip into the pocket to form π-π/cation-π interactions and hydrogen bonds with H373, and hydrogen bonds with UDP. We propose that the additional S371 residue could increase flexibility of the loop, thus enables R368 to interact with H373 for the binding of UDP-Api. When these amino acids were mutated to alanine, the activity of GuApiGT was decreased (Supplementary Fig. 87). We further simulated the structures of GuApiGT/UDP-Xyl/**2** and GuApiGT/UDP-Glc/**2** (Supplementary Fig. 88). The configuration of Glc in the complex structure is unreasonable due to a twist conformation between boat and chair, whereas 6-OH could suppress the attack of UDP-sugar C1‴ to 2″-OH of **2** (Supplementary Fig. 89). For the binding of Xyl, the MM/GBSA (molecular mechanics, the generalized Born model and solvent accessibility) binding free energy of GuApiGT/UDP-Xyl/**2** is higher than that of GuApiGT/UDP-Api/**2** (Supplementary Fig. 90)[32]. These results were consistent with our observation that GuApiGT could not accept UDP-Glc or UDP-Xyl.

For inverting GTs, the distance between the hydroxyl oxygen atom of sugar receptor (O2″) and C1‴ of UDP-sugar, as well as the angle of O2″, C1‴, and the UDP oxygen atom next to C1‴ (O3), are critical for the glycosylation reaction[13]. We performed 500-ns well-tempered metadynamic simulations using the distance (CV1: O2″-C1‴) and angle (CV2: O2″-C1‴-O3) as the first and second collective variables (CV), respectively[33]. Optimal conformations should have CV1 lower than 4.5 Å and CV2 greater than 90°. As shown in Fig. 4f, only UDP-Api, but not UDP-Glc or UDP-Xyl, exhibited reasonable local minima of free energy surfaces (FES).

We further conducted QM/MM calculations using the ONIOM method implemented in Gaussian 16 (rev C.01) to derive the transition states[34]. CV1/CV2 values in the optimized transition state (TS) structure is 2.1 Å/152.1° (Fig. 4g, Supplementary Data 2-4). The activation barrier and related product energy for the transfer of apiose is 14.5 and -1.3 kcal/mol, respectively (Fig. 4h). During the process to form the glycosidic bond between O2″ of **2** and C1‴ of UDP-Api, H18 could partially deprotonate **2**, with the assistance of D115. Once the reaction is completed, D115 is protonated in the product complex. On the other hand, when we remove the atomic charge of outer MM region residues, R368 and E272 could notably impact the activation barrier, with ΔΔE of 12.06 and 21.67 kcal/mol, respectively (Supplementary Table 6 and Supplementary Fig. 91).

**Protein engineering of GuApiGT to change its sugar donor selectivity**
To verify the role of RLGSDH motif on sugar donor selectivity of GuApiGT, we conducted site-directed mutagenesis. First, we deleted the additional S371, and the catalytic activity was decreased (Fig. 5a).

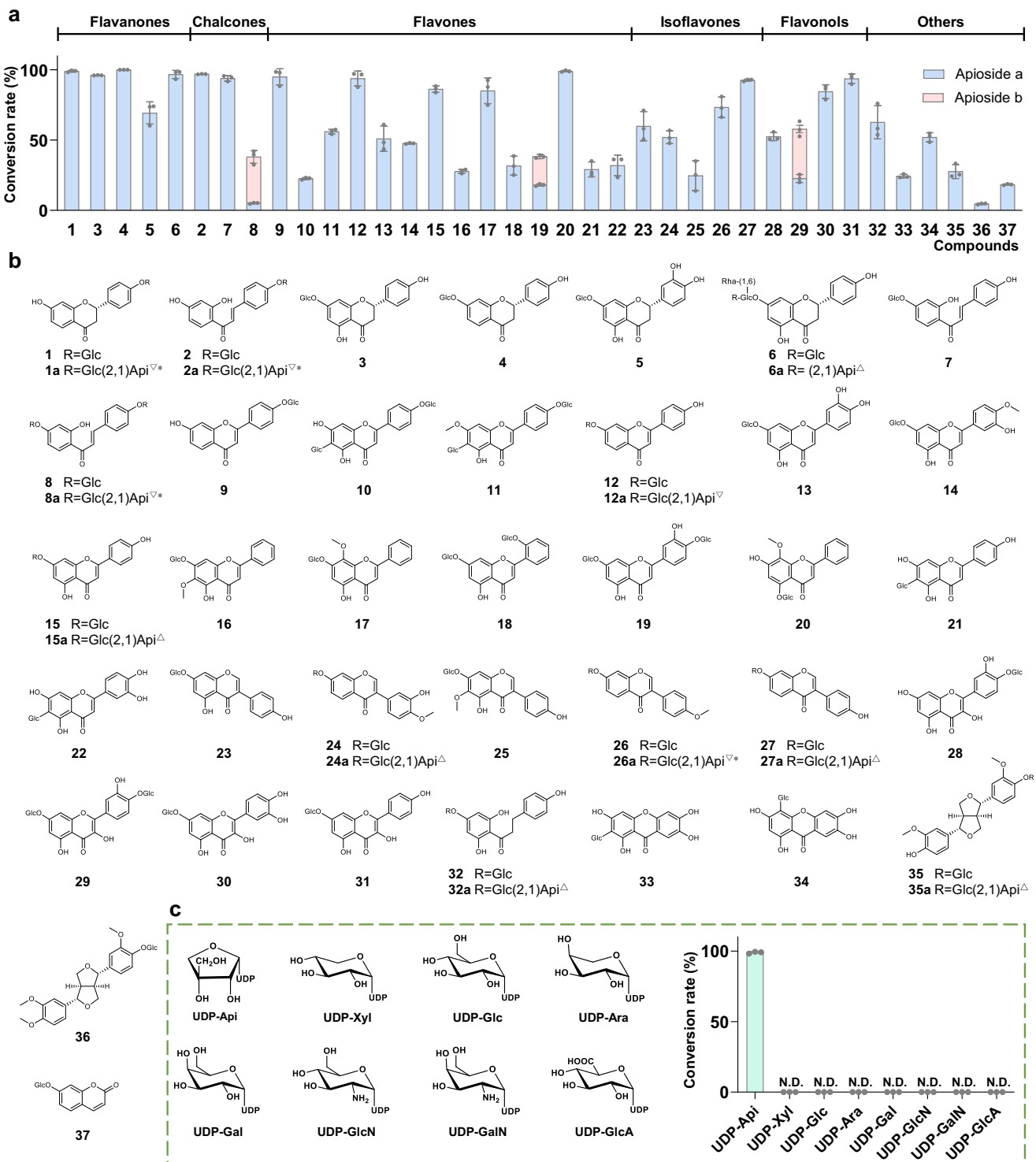

**Fig. 3 | Substrate promiscuity of GuApiGT. a** Conversion rates of apiosylated products of different types of substrates catalyzed by GuApiGT. **b** Structures of substrates **1-37** and part of the products. The regular triangles represent products purified and identified by NMR; The inverted triangles represent compounds identified by comparing with reference standards; The asterisks represent endogenous compounds of *G. uralensis*. **c** Structures of sugar donors and the conversion rates. **1** was used as sugar acceptor. Data are presented as mean values ± SD (*n* = 3 biologically independent samples) (**3a** and **c**). The source data underlying Fig. (3a and c) are provided in a Source Data file.

Then we replaced H373 with glutamine, which is usually the last residue of PSPG box of UDP-Glc-preferring UGTs, as glutamine could form hydrogen bonds with 2-OH and 3-OH of glucose. The H373Q mutant also showed decreased activity. Interestingly, the S371/H373Q mutant could accept UDP-Xyl as sugar donor (Fig. 5a, b and Supplementary Figs. 92, 93). Similarly, we deleted R368, L369, and G370, respectively.

All the double mutants could accept UDP-Xyl, and L369/H373Q was the most active one, with a conversion rate of almost 100%.

The structures of glucose and xylose differ in the $CH_2OH$ substituent at C-5. In our previous report, T145 in GgCGT is a key amino acid to form hydrogen bond with 6-OH of UDP-Glc[26]. T145 is mapped to Ile136 in GuApiGT (Supplementary Fig. 94). Thus, we continued to

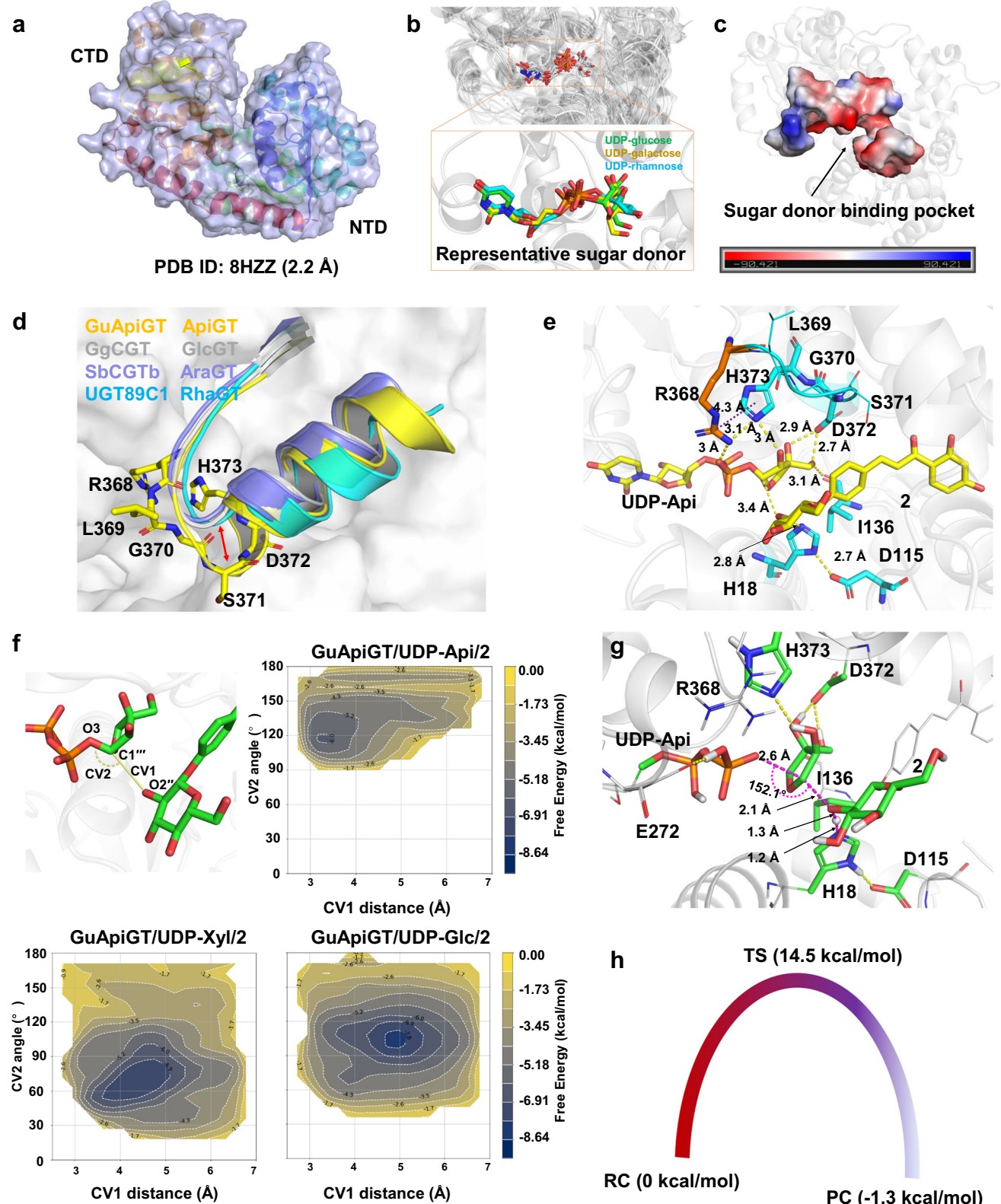

construct a series of triple mutants. I136T/G370/H373Q showed the highest catalytic activity to accept UDP-Glc (Fig. 5a, b and Supplementary Fig. 95). The co-incubation of UDP-Xyl and UDP-Glc further confirmed the significance of T136 on the preference towards UDP-Glc (Supplementary Fig. 96).

The L369/H373Q and I136T/G370/H373Q mutants could catalyze a series of substrates using UDP-Xyl and UDP-Glc as sugar donor,

respectively (Fig. 5c and Supplementary Figs. 97–122). Products **32b** and **3c** were purified from scaled-up reactions and their structures were identified by NMR analysis as 2″-*O*-xyloside of trilobatin and 2″-*O*-glucoside of naringenin, respectively (Supplementary Figs. 123–132).

We further employed hydrogen-deuterium exchange mass spectrometry (HDX-MS) to elucidate the protein conformation of GuApiGT, and L369/H373Q and I136T/L369/H373Q mutants in the solution

**Fig. 4 | Structural basis for the UDP-apiose selectivity of GuApiGT. a** The crystal structure of GuApiGT. **b** Superimposition of 26 plant crystal structures (the sugar donors are highlighted). **c** The sugar donor binding pocket of GuApiGT. **d** The sugar binding region in crystal structures of representative ApiGT, GlcGT (GgCGT, PDB ID: 6L5P), AraGT (SbCGTb, PDB ID: 6LFZ), and RhaGT (UGT89C1, PDB ID: 6IJA). The RLGSDH motif in GuApiGT is highlighted as yellow sticks. **e** A representative configuration of GuApiGT/UDP-Api/**2** extracted from MD simulations. The hydrogen-bond interactions and π-π/cation-π interactions are shown as yellow and purple dashes, respectively. The key amino acids interacted with ligands are highlighted using sticks. The unique R368 is depicted as orange sticks, the others as blue. The other amino acids in key motif are depicted using lines. **f** Metadynamic simulations of GuApiGT with different sugar donors (Api, Xyl, and Glc). CV1, the distance of O2′′-C1′′′ (Å); CV2, the angle of O2′′-C1′′′-O3 (°). **g** QM/MM optimized geometry of the transition state (TS) at the theory of B3LYP/6-311++G(2d,2p):AMBER with the electronic embedding scheme and thermal zero-point energy calculated from the theory of B3LYP/6-31G(d):AMBER. The QM region atoms, hydrogen bonds, and key angle and distances are highlighted in green sticks, yellow dashes, and magenta dashes, respectively. The MM region atoms are depicted using lines. **h** An energy profile for the apiosylation of GuApiGT. RC, reactant complex; TS, transition state; PC, product complex. The source data underlying Fig. (4f) are provided in a Source Data file.

state[35,36]. The peptide coverage was 90.1% (Supplementary Figs. 133, 134). Compared with the wild type, peptide L363-D372 of the mutants showed decreased deuterium uptake, indicating the R[368]L[369]G[370]S[371] loop became compact and rigid after mutagenesis, and thus may be able to interact with xylose or glucose (Fig. 5d, e and Supplementary Figs. 135, 136). The L135-E156 peptide of I136T/L369/H373Q mutant exhibited noticeable increase of deuterium uptake, verifying the significance of T136 in recognizing UDP-Glc (Supplementary Fig.137). For the PSPG box, the R316-M334 and I335-F344 peptides only showed minor changes, while I345-Q362 and L363-D372 changed significantly upon mutagenesis. This result indicated the R316-F344 and I345-H373 parts were responsible for the binding with UDP and the sugar moiety, respectively.

The recognition of UDP-Xyl and UDP-Glc was also supported by thermal shift assay[37]. Compared to the WT, the melting temperature ($T_m$) of the mutants increased by 1.2-5.4°C and 0.9-2.7°C, respectively, when co-incubated with UDP-Xyl or UDP-Glc (Fig. 5f). These results proved that UDP-Xyl and UDP-Glc could bind with the mutant protein and increase stability. We simulated the structure models of L369/H373Q mutant/UDP-Xyl/**2** and I136T/L369/H373Q mutant/UDP-Glc/**2** complexes (Supplementary Fig. 138a). The binding free energies and local minima from metadynamic simulations were consistent with the experimental results (Supplementary Fig. 138b, c). Moreover, the volume of sugar binding pocket decreased upon mutagenesis (Supplementary Fig. 138d).

### Protein engineering of Sb3GT1 to gain apiosylation activity

To further prove the critical role of the RLGSDH motif in UDP-Api selectivity, we conducted site-directed mutagenesis of Sb3GT1. Sb3GT1 is an efficient plant flavonoid 3-O-glycosyltransferase which could accept at least five sugar donors except for UDP-Api[38]. It shares 19.8% amino acid sequence identity with GuApiGT (Supplementary Fig. 139). We solved the complex crystal structure of Sb3GT1/UDP at 1.9 Å resolution (Fig. 6a and Supplementary Table 7), and the RMSD compared with GuApiGT is 2.64 Å. The RLGSDH (368-373) motif in GuApiGT is mapped to FFGDQ (372-376) of Sb3GT1.

Based on structural analysis, we inserted a serine residue into the motif and constructed the 375S/Q377H mutant of Sb3GT1, as well as the F372R/Q376H and F372R/375S/Q377H mutants. All the three mutants could catalyze kaempferol (**66**) into its 3-O-apioside, according to the characteristic [Y⁰-H]⁻ ion at $m/z$ 284 in LC/MS analysis (Fig. 6b and Supplementary Fig. 140)[38]. We further solved the crystal structure of Sb3GT1 375S/Q377H mutant in complex with UDP-Glc at 1.43 Å resolution (Fig. 6c and Supplementary Table 7). The motif structure of the mutant was larger than that of WT, which may be critical for the sugar donor selectivity towards UDP-Api. Interestingly, GuApiGT could not accept free flavonoids or 3-O-glycosides, which could be interpreted by molecular docking and MM/GBSA binding free energy calculations (Supplementary Fig. 141).

### The RLGSDH motif is general for leguminosae plants

To discover more apiosyltransferases, we analyzed the online plant transcriptome databases[39]. A total of 121 candidate genes were discovered from 39 plant species, using the unique 45-amino acid PSPG box as a filter (Supplementary Table 8). Interestingly, all the species belong to Leguminosae family. These genes were closely clustered with GuApiGT in the phylogenetic tree, except for three genes clustered with ZjOGT38 and TcOGT4 (Fig. 7a).

Majority of these ApiGT genes contain the RLGSDH motif in the PSPG box (Fig. 7b). Among the 39 plants, *P. thomsonii*, *S. suberectus*, *G. glabra*, and *G. inflata* had been reported to contain flavonoid apiosides[14,40,41]. Thus, we cloned PtApiGT, SsApiGT, GgApiGT, and GiApiGT from the plants, and identified them as apiosyltransferases by enzyme catalysis reactions (Fig. 7c and Supplementary Fig. 142). Their amino acid sequences were highly conservative, with identity of 90.07% (Supplementary Fig. 143). Very recently, Reed et al. reported UGT73CY2 as a triterpenoid apiosyltransferase, which has a PSPG box of 44 amino acids[16]. After submission of the present work, Yamashita et al. reported UGT94AX1 from *Apium graveolens* (Apiaceae family), which also contains a 44-amino acid PSPG box[42]. Their amino acid sequence identity with GuApiGT was 21% and 23%, respectively. Thus, the unique 45-amino acid PSPG box and the RLGSDH motif may be general for apiosyltransferases from Leguminosae plants.

### De novo biosynthesis of flavonoid apiosides in tobacco

Liquiritin apioside (**1a**) and isoliquiritin apioside (**2a**) are important bioactive compounds in the Chinese herbal medicine Gan-Cao (licorice)[10]. Thus far, they could only be prepared by purification from licorice, which needs to grow for at least 3-4 years. The discovery of GuApiGT paved the way for their de novo biosynthesis. While *E. coli* and yeast are widely used as chassis for de novo biosynthesis of natural products, the yields for flavonoids are usually low[43,44]. Thus far, the most productive engineering system for flavonoid glycosides had a yield of 100 mg/L. *Nicotiana benthamiana* (tobacco) is a rapid growing and high biomass plant[45,46], and may be a suitable host for the production of flavonoids. The de novo tobacco biosynthesis of several important natural products has been achieved, including taxadiene-5α-ol, colchicine, and (-)-deoxypodophyllotoxin[47–50].

To evaluate the suitability of *N. benthamiana* as a potential platform for the production of apiosides (Fig. 8a), *Agrobacterium*-mediated transient expression of *GuApiGT* was performed using pEAQ-HT-DEST1 with a 35 S promoter[51]. *UAXS* was co-infiltrated into tobacco with *GuApiGT* to supplement the UDP-apiose donor. Isoliquiritin (**2**) and UDP-GlcA were infiltrated into the leaves after *GuApiGT* and *UAXS* expression for 3 days. Leaf discs from the infiltrated parts were sampled 4 days post-infiltration. The samples were extracted and analyzed by LC/MS. Product **2a** could be detected at noticeable amounts (Supplementary Fig. 144). To optimize the agrobacterium strain, pEAQ-HT-DEST1-*GuApiGT* was transferred to five *Agrobacterium* strains, including AGL1, GV3101, C85C1, LBA4404, and GV2260. Among the strains, GV2260 showed the highest conversion and was selected as the most suitable strain for the expression of GuApiGT (Fig. 8b).

For the de novo biosynthesis of (iso)liquiritin apiosides, we designed 3 modules, including the flavonoid aglycone module (module 1), UDP-donor module (module 2), and glycosyltransferase module (module 3). For module 1, AtPAL, AtC4H, At4CL, AtCHS and GuCHR

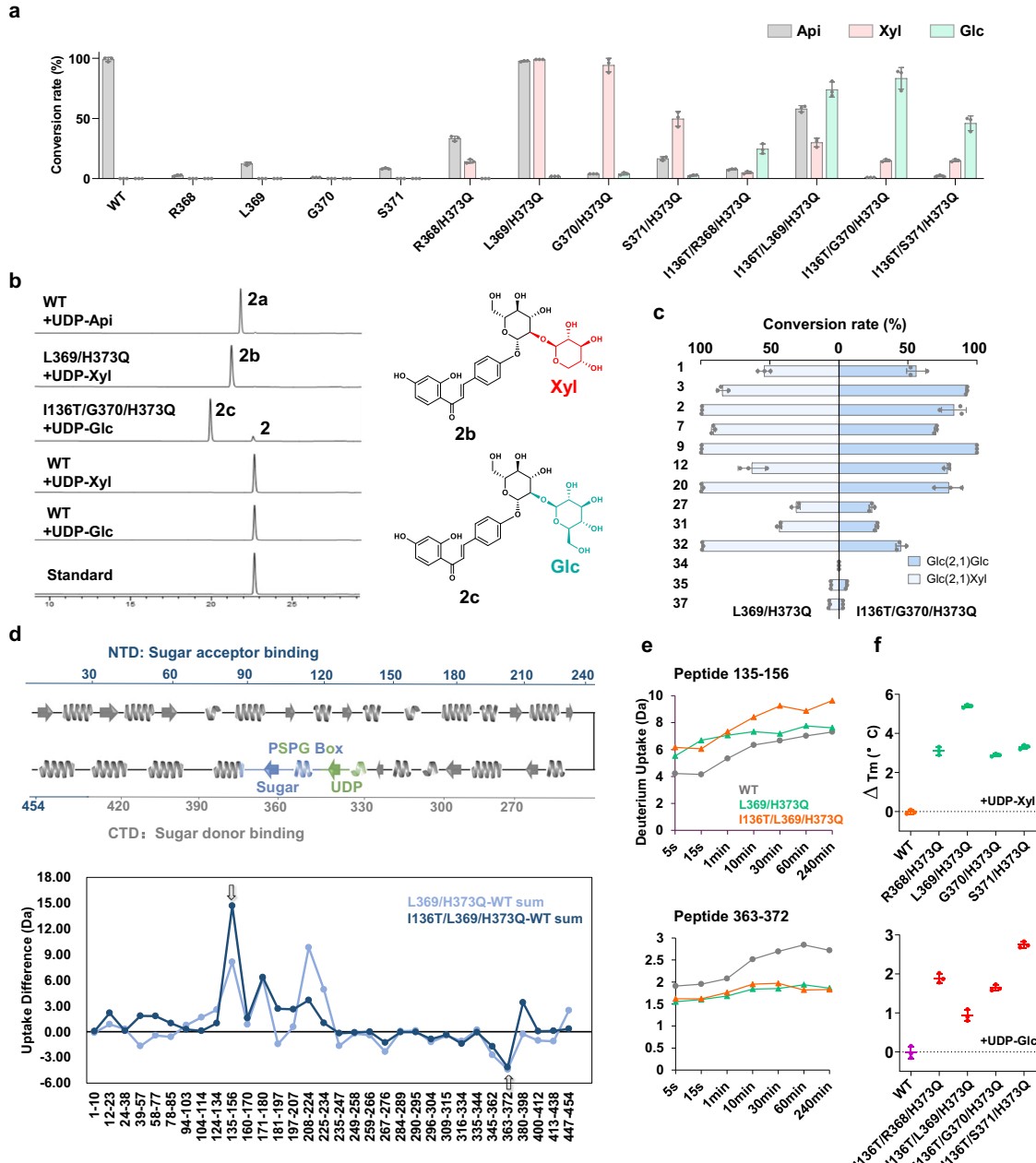

**Fig. 5 | Alteration of sugar donor selectivity of GuApiGT mutants. a** The glycosylation conversion rates of the wild type (WT) and mutants, using **2** as sugar acceptor, and UDP-Api, UDP-Xyl, or UDP-Glc as sugar donor. **b** HPLC chromatograms of enzyme catalytic products. **c** Substrate promiscuity of GuApiGT mutants L369/H373Q and I136T/G370/H373Q. For substrate structures, see Fig. 3. **d** Deuterium uptake differences of L369/H373Q-WT and I136T/L369/H373Q-WT for all peptides in the HDX-MS experiments, calculated as the sum of all time points. The enzyme secondary structure and the PSPG box is shown on the top. **e** Deuterium uptake plots of peptides 135-156 and 363-372 at different time points. **f** Protein thermal shift assay measuring the changes in melting temperature ($\Delta T_m$) of WT and mutants. Top, co-incubated with UDP-Xyl; bottom, co-incubated with UDP-Glc. Data are presented as mean values ± SD ($n = 3$ biologically independent samples) (**5a, c** and **f**). The source data underlying Fig. (5a, c–f) are provided in a Source Data file.

were used to synthesize isoliquiritigenin (**2′**). Pgm, GalU, CalS8 and UAXS were used for module 2 to produce UDP-Glc and UDP-Api. For module 3, GuApiGT and the previously reported GuGT14 were used as glycosyltransferases[52]. However, **7a** was generated as a major byproduct when GuGT14 was used. It was due to the poor regio-selectivity of GuGT14 and endogenous glycosyltransferases from tobacco (Fig. 8c and Supplementary Fig. 145). Then we discovered *GuGT53* from *G. uralensis*, which showed a similar expression pattern as *GuCHS*, *GuCHR*, and *GuApiGT* (Fig. 1d, e). GuGT53 (UGT88E28, GenBank accession number OQ266890) could regio-selectively and efficiently catalyze 4′/4-*O*-glycosylation of liquiritigenin (**1′**) and **2′** into liquiritin

(**1**) and isoliquiritin (**2**), respectively (Supplementary Fig. 146). The yield of isoliquiritin apioside was improved when GuGT14 was replaced by GuGT53. When module 1 or 2 was absent, no **2a** could be generated. It was generated after UDP-GlcA or **2′** was injected into the tobacco leaves (Fig. 8c). Moreover, we cloned *GuCHI* from *G. uralensis* to replace *AtCHI*, as **1a** could not be detected when *AtCHI* was used in module 1 (Supplementary Fig. 147). The OD600 of *Agrobacterium* was also optimized, and OD600 0.2 for each gene was found to be the most efficient concentration for **1a** (Fig. 8d).

Finally, the contents of **1a** and **2a** in tobacco leaves were 5.46 and 4.73 mg/g (dry weight, DW), respectively, with the above optimized

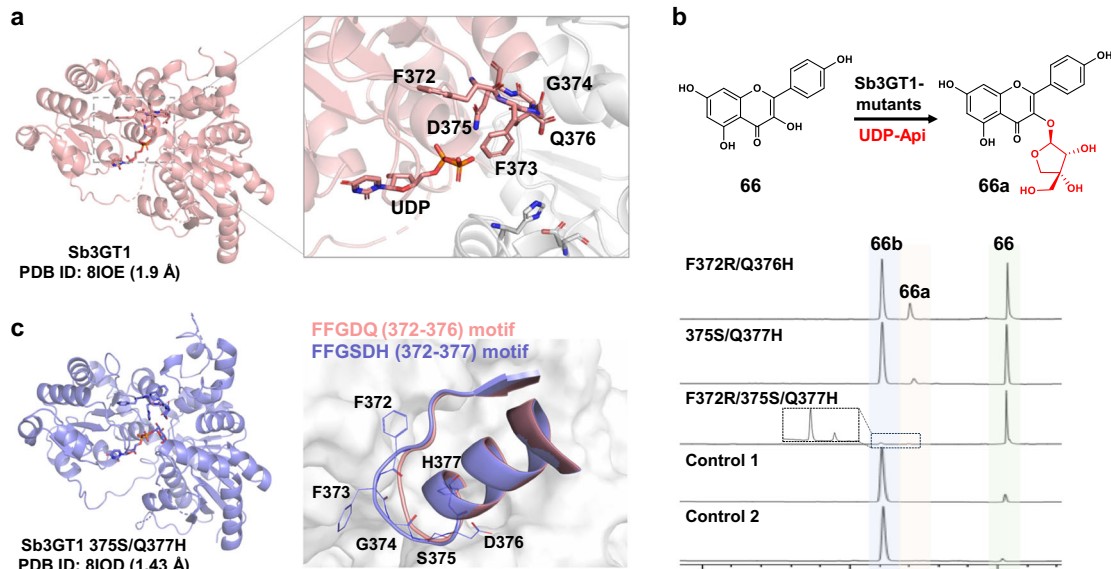

**Fig. 6 | Apiosylation activity of Sb3GT1 mutants. a** The crystal structure of Sb3GT1 (PDB ID: 8IOE) and the FFGDQ (372-376) motif. The image on the right is an enlargement of the dashed rectangle, where the red part represents CTD and the grey part represents NTD. The amino acids in key motif of Sb3GT1 are highlighted using sticks. **b** Functional characterization of Sb3GT1 and its mutants. Control 1, the acceptor was incubated with Sb3GT1 and an UDP-Api supply system (UDP-GlcA, UAXS, NAD⁺). Control 2, the acceptor was incubated with Sb3GT1 and UDP-Xyl. Proposed structures for **66a**, kaempferol 3-*O*-apioside; **66b**, kaempferol 3-*O*-xyloside. **c** The crystal structure of Sb3GT1 375S/Q377H mutant (PDB ID: 8IOD) and superimposition of its key motif to that in wild type. The amino acids in key motif of 375S/Q377H mutant are depicted using lines.

conditions (Fig. 8e, f). By using different gene combinations for module 1, we realized the de novo biosynthesis of eight more flavonoid apiosides. The basic skeleton could be flavanone, chalcone, or flavone, and the yields ranged from 0.19-6.25 mg/g (DW) (Supplementary Figs. 148–155).

In conclusion, we identified the missing phenolic apiosyltransferase GuApiGT from *G. uralensis*. GuApiGT could efficiently and regio-selectively catalyze 2″-*O*-apiosylation of flavonoid glycosides, and showed strict sugar donor selectivity towards UDP-Api. This selectivity was highly related with the unique 45-amino acid PSPG box and the key RLGSDH sugar binding motif. Through theoretical calculations and rational design, we altered the sugar donor selectivity of GuApiGT and Sb3GT1. The 45-amino acid PSPG box and the RLGSDH motif may be general for Leguminosae plants, and helped to discover 4 other apiosyltransferases. We also achieved de novo biosynthesis of at least 10 flavonoid apiosides in tobacco, and the yields could be up to around 6 mg/g. This work realized efficient biosynthesis of flavonoid apiosides, including the important bioactive natural product liquiritin apioside. It also highlights the sugar donor selectivity mechanisms of GuApiGT, and sets a good example for functional evolution and protein engineering of catalytic enzymes.

## Methods
### Plant materials
The fresh plant of *Glycyrrhiza uralensis* Fisch. (2-3 years old) was collected from Inner Mongolia Autonomous Region of China in August 2019 for total RNA extraction and transcriptome sequencing. The seeds of *G. glabra* and *G. inflata* were obtained from Gan-Su (China) and were sown in our laboratory under natural conditions. To extract RNA, 3-week-old seedlings were used. The fresh plant of *Pueraria thomsonii* (1-2 years old) was collected from Anhui Province of China in June 2022 for total RNA extraction.

### Total RNA isolation and transcriptome sequencing
The total RNA was extracted using the TranZol™ kit (Transgen Biotech, China) following the manufacturer's instructions, and was used to synthesize the first-stranded complementary DNA (cDNA) using TransScript one-step genomic DNA (gDNA) removal and cDNA synthesis SuperMix (Transgen Biotech, China). The transcriptome data of different parts of *G. uralensis* were acquired using Illumina sequencing platform by Majorbio Bioinformatics Technology Co., Ltd (Shanghai, China).

### Bioinformatics
Co-expression analysis was conducted using R studio. Genes highly expressed in the roots (fragments per kilobase of transcript per million mapped reads (FPKM) ≥ 20 in two biological replicates) were selected for co-expression analysis. *GuCHS* and *GuCHR* were used as 'bait'. The co-expressed genes were further filtered by Pfam (https://www.ebi.ac.uk/interpro), NR (non-redundant protein sequences, https://www.ncbi.nlm.nih.gov/), and UniProtKB/Swiss-Prot databases (https://www.uniprot.org/) annotation and Spearman's correlation coefficient (r ≥ 0.8). The analysis was performed using *G. uralensis* RNA-seq transcriptome data from different tissues. The co-expression network was visualized by Cytoscape.

Homologous plant ApiGT genes were searched using online transcriptome data via China National GeneBank (https://db.cngb.org/blast/) which contains 1,000 plants project, transcriptome shotgun assembly proteins, non-redundant protein sequences, and UniProtKB/Swiss-Prot databases. GuApiGT was used as the query sequence. BLASTP was used for BLAST search with default parameters. Molecular phylogenetic analysis was conducted using MEGA6 software with the maximum likelihood method. The bootstrap consensus tree inferred from 1,000 replicates was taken to represent the evolutionary history of the taxa analyzed.

### Molecular cloning, site-directed mutagenesis, and expression of GTs
The full-length GT genes were amplified from cDNA using TransStart® FastPfu DNA Polymerase (Transgen, China) and were cloned into pET-28a(+) vector (Invitrogen, USA) by the Quick-change method. Mutants were constructed using a Fast Mutagenesis System kit (Transgen

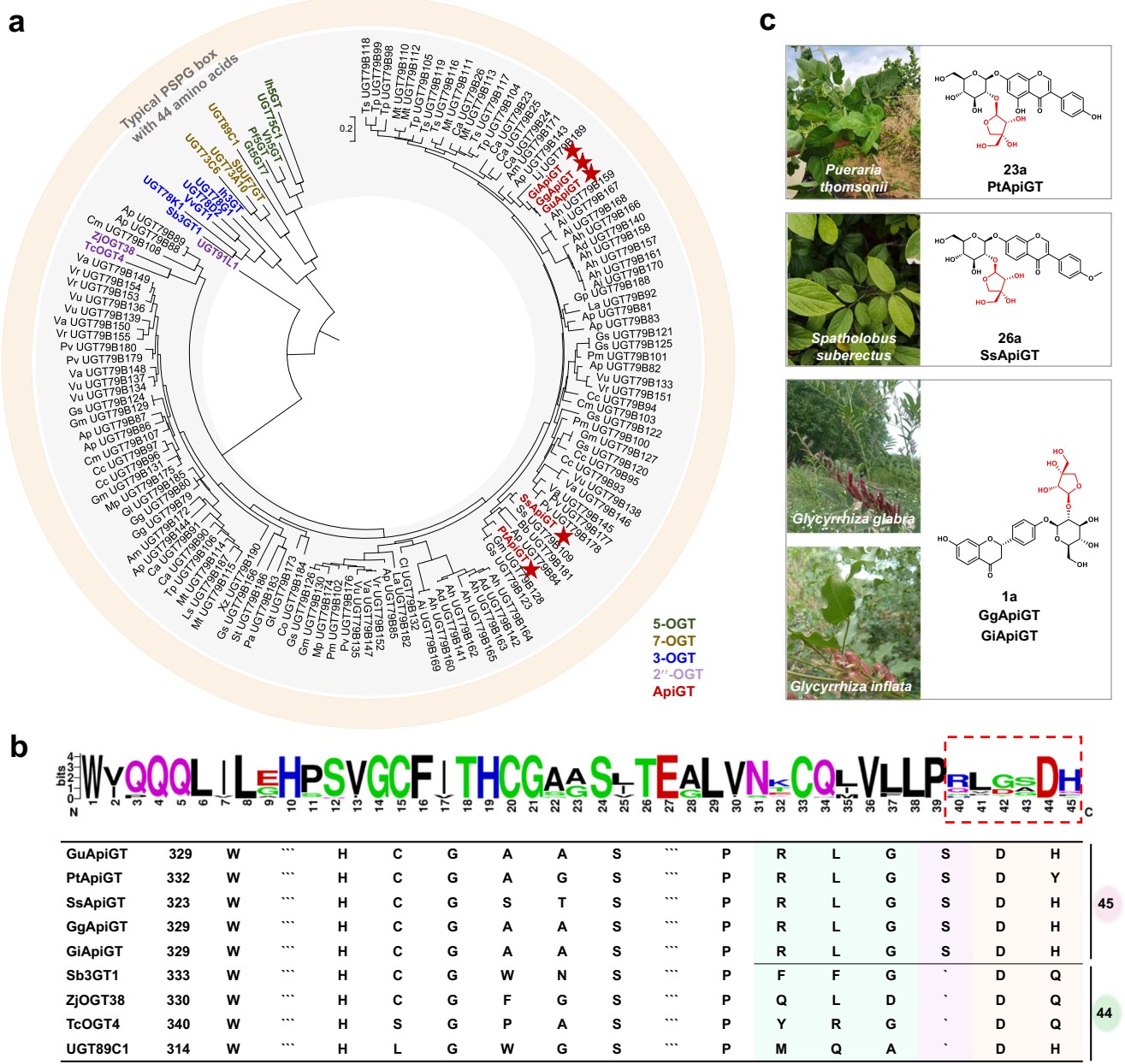

**Fig. 7 | Discovery of apiosyltransferases from Leguminosae plants.**
**a** Phylogenetic analysis of 121 potential apiosyltransferase genes discovered from 39 plants and 18 reported flavonoid UGTs. **b** Sequence alignment of PSPG box of the 121 apiosyltransferases. The red dashed box shows the unique RLGSDH motif.

The image was created using WebLogo (https://weblogo.berkeley.edu). The table listed conserved amino acids of 9 UGTs, including the 5 characterized ApiGTs. **c** The functions of PtApiGT (*Pueraria thomsonii*), SsApiGT (*Spatholobus suberectus*), GgApiGT (*Glycyrrhiza glabra*), and GiApiGT (*Glycyrrhiza inflata*).

Biotech, China) according to the manufacturer's instructions. The primers are given in Supplementary Data 1. The full length of *SsApiGT* was synthesized by Tsingke Biological Technology Incorporation (Beijing, China). The recombinant plasmid pET-28a(+)-GTs were introduced into *E. coli* BL21(DE3) (Transgen Biotech, China) for heterologous expression. Single colonies were incubated in LB media (50 μg/mL kan$^+$) on a rotary shaker at 37 °C. When the $OD_{600}$ value was around 0.6, protein expression was induced with 0.1 mM IPTG for 20 h at 18 °C. The cell pellets were collected by centrifugation (6408 × $g$ for 10 min at 4 °C). Then the cells were resuspended in 15 mL of lysis buffer (10 mM imidazole, 20 mM Tris, 200 mM NaCl, 2% glycerol ($v/v$), pH 7.4) and ruptured by sonication on ice for 15 min. The cell debris was removed by centrifugation at 14,420 × $g$ for 45 min at 4 °C. The recombinant proteins were purified using a nickel-affinity column. The elution buffer included two types: one containing 30 mM imidazole,

20 mM Tris, 200 mM NaCl and 2% glycerol ($v/v$) to elute impurities, and the other containing 300 mM imidazole, 20 mM Tris, 200 mM NaCl and 2% glycerol ($v/v$) to elute the target protein. All the buffers were adjusted to pH 7.4 by HCl. The impurities were eluted with 50 mL elution buffer containing 30 mM imidazole. Then, GuApiGT recombinant protein was eluted by 20 mL elution buffer containing 300 mM imidazole. The protein purity was analyzed by SDS-PAGE (Supplementary Fig. 2). The purified protein was concentrated and desalted by a 30 kDa ultrafiltration tube (Merck Millipore) with a storage buffer (20 mM Tris, 200 mM NaCl, 20% ($v/v$) glycerol, pH 7.4). The processes for the other GTs were the same as that of GuApiGT.

### Enzyme activity assay
The reactions were carried out in 100-μL systems containing 50 mM $NaH_2PO_4$-$Na_2HPO_4$ (pH 8.0), 0.1 mM sugar acceptor, 6 mM NAD$^+$,

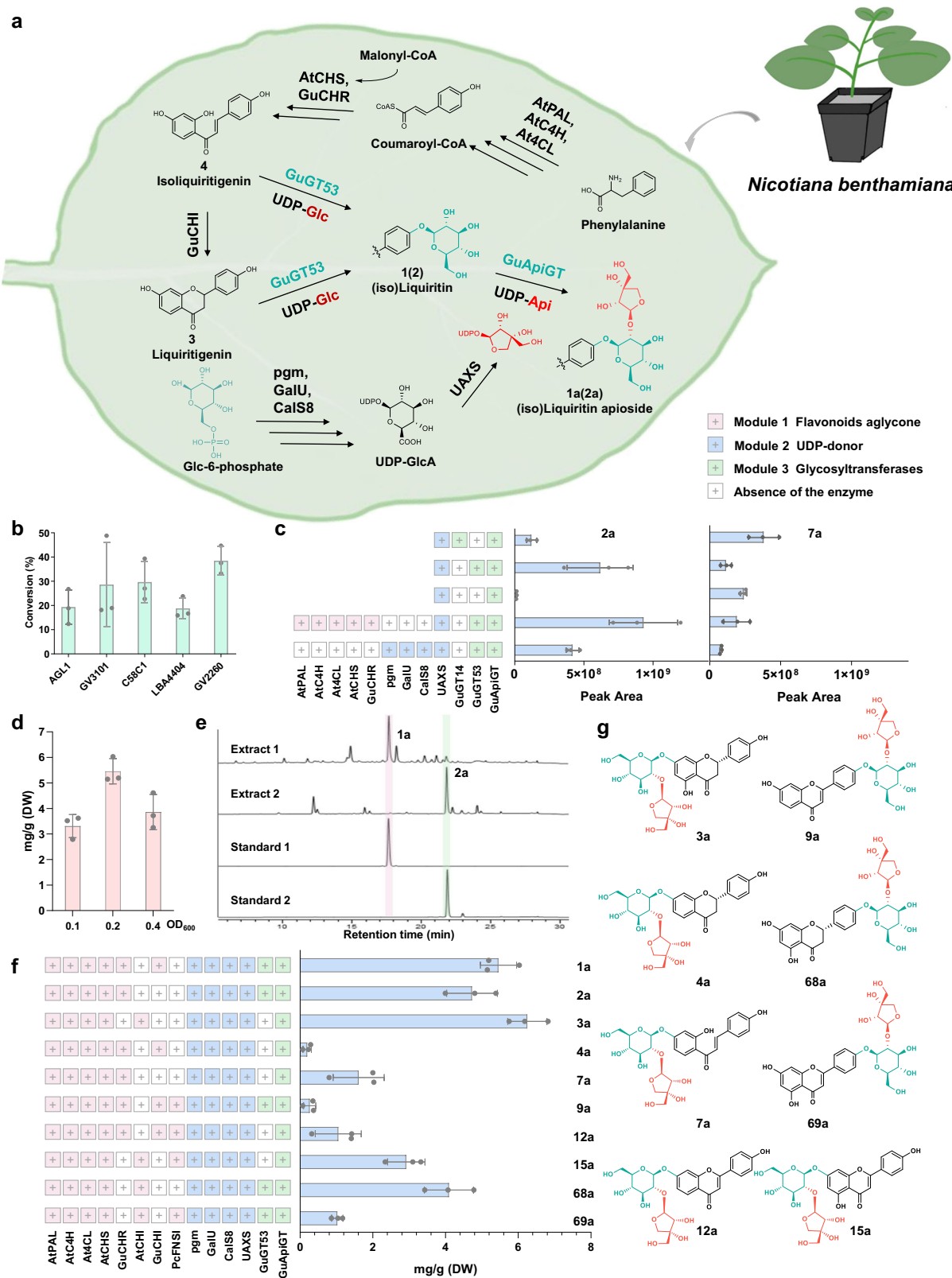

1.5 mM UDP-GlcA, 50 µg of UAXS, and 5 µg of purified GuApiGT or mutants at 37 °C for 3 h. For other sugar donors, the reactions were carried out in 100-µL systems containing 50 mM NaH₂PO₄-Na₂HPO₄ (pH 8.0), 0.1 mM sugar acceptor, 0.5 mM UDP-Xyl or 0.5 mM UDP-Glc, and 40 µg of purified mutants at 37 °C for 3 h. For Sb3GT1 and its mutants, the reactions were carried out in 100-µL systems containing 50 mM Tris-HCl (pH 9.0), 0.05 mM sugar acceptor, 6 mM NAD⁺, 1.5 mM

UDP-GlcA, 50 µg of UAXS, and 100 µg of purified Sb3GT1 or mutants at 45 °C for 3 h. The reactions were terminated by adding 200 µL pre-cooled methanol and then centrifuged at 21,130 × g for 20 min. The supernatants were filtered through a 0.22-µm membrane and then analyzed by LC/MS. The samples were separated on an Agilent Zorbax SB-C18 column (4.6 × 250 mm, 5 µm) at a flow rate of 1 mL/min at room temperature. The mobile phase was a gradient elution of solvents A

**Fig. 8 | De novo biosynthesis of (iso)liquiritin apioside and analogues in tobacco. a** The engineered biosynthetic pathway of **1a** and **2a** in *Nicotiana benthamiana*. **b**–**d** Optimization of *Agrobacterium* strains, bioparts, and optical density of bacterial cultures for the biosynthesis of **1a** and **2a**, respectively. Three days after infiltration, isoliquiritin and UDP-GlcA were supplemented for lines 1-3; UDP-GlcA and isoliquiritigenin were supplemented for lines 4 and 5, respectively. **e** HPLC chromatograms of engineered tobacco extracts. **Extract 1** and **Extract 2** represent samples to produce **1a** and **2a**, respectively. **Standard 1** and **Standard 2** were

reference standards of liquiritin apioside and isoliquiritin apioside, respectively. **f, g** De novo biosynthesis of **1a**, **2a**, and 8 other flavonoid apiosides. The structures of **1a**, **2a**, **12a** and **15a** were identified by comparing with reference standards; **3a, 4a, 7a** and **9a** were identified by comparing with catalytic products of GuApiGT; and **68a** and **69a** were characterized by analyzing mass spectral data. Data are presented as mean values ± SD ($n = 3$ biologically independent samples) (**8b, c, d, f**). The source data underlying Fig. (8b, c, d and f) are provided in a Source Data file.

(water containing 0.1% formic acid) and B (acetonitrile, ACN), and the gradient programs were listed in Supplementary Table 3. The conversion rates in percentage were calculated from peak areas of glycosylated products and sugar acceptors in HPLC/UV chromatograms (Agilent 1260, USA). MS analysis was performed on a Q-Exactive hybrid quadrupole-Orbitrap mass spectrometer equipped with a heated ESI source (Thermo Fisher Scientific, USA). The MS parameters were as follows: sheath gas pressure 45 arb, aux gas pressure 10 arb, discharge voltage 4.5 kV, capillary temperature 350 °C. $MS^1$ resolution was set as 70,000 FWHM, AGC target 1*E6, maximum injection time 50 ms, and scan range $m/z$ 100-1000. $MS^2$ resolution was set as 17,500 FWHM, AGC target 1*E5, maximum injection time 100 ms, NCE 35. The mass spectra were recorded in the negative ion mode for all the substrates except for **19** and **29**.

### Biochemical properties of GuApiGT

To determine the optimal reaction time, 9 time points between 5 and 600 min were tested. To optimize the pH value, different reaction buffers with pH from 3.0-6.0 (citric acid-sodium citrate buffer), 6.0-8.0 ($Na_2HPO_4$-$NaH_2PO_4$ buffer), 7.0-8.5 (Tris-HCl buffer), and 9.0-10.8 ($Na_2CO_3$-$NaHCO_3$ buffer) were tested. To optimize the reaction temperature, the reactions were incubated at different temperatures (4, 18, 25, 30, 37, 45, 60 °C). To determine the effects of divalent metal ions on enzyme activities, EDTA, $BaCl_2$, $CaCl_2$, $FeCl_2$, $MgCl_2$, $ZnCl_2$ and $CuCl_2$ were added individually at a final concentration of 5 mM (Supplementary Fig. 4). All enzymatic reactions (100 µL reaction mixtures including 0.1 mM isoliquiritin, 6 mM NAD+, 1.5 mM UDP-GlcA, 50 µg of UAXS, and 2 µg of purified GuApiGT) were conducted in three parallel experiments ($n = 3$). The reactions were terminated with pre-cooled methanol and centrifuged at 21,130 × $g$ for 20 min for HPLC analysis as described above.

### Preparation of UDP-apiose

The reaction mixtures contained 100 µL buffer (100 mM triethylamine phosphate, pH 8.0), 0.1 mM NAD+, 10 mM UDP-GlcA, and 0.48 mg of UAXS. A total of 30 parallel tubes were used. The reactions were performed at 25 °C for 4 h and then centrifuged at 21,130 × $g$ for 30 min. The products were subsequently purified by reversed-phase HPLC. HPLC was performed on an Inertsustain AQ-C18 column (5 µm, 4.6 × 250 mm; GL Sciences, Tokyo, Japan) at a flow rate of 1.0 mL/min. The mobile phase was a gradient elution of solvents A (100 mM *N,N*-dimethylcyclohexylamine phosphate buffer, pH 6.5) and B (30% (*v/v*) ACN). A gradient elution program was used: 0 min, 100% A; 13 min, 100% A; 35 min, 33% A; 39 min, 33% A; 40 min, 100% A. The eluted fractions were monitored by measuring the UV absorbance at 262 nm (Supplementary Fig. 156). After freeze-drying, UDP-apiose was dissolved with triethylamine phosphate for use.

### Determination of GuApiGT kinetic parameters

In a final volume of 25 µL with 50 mM $Na_2HPO_4$-$NaH_2PO_4$ buffer (pH 8.0), 2 µg/mL protein, 480 µmol/L of saturated UDP-apiose, and different concentrations of compound **2** (1, 2.5, 5, 10, 30, 40, 60, 80, 150 µmol/L) were added. The reactions were quenched with pre-cooled methanol after incubating at 37 °C for 15 min, and then centrifuged at 21,130 × $g$ for 15 min. The supernatants were used for HPLC analysis. All

experiments were performed in triplicate. The conversion rates in percentage were calculated from HPLC peak areas of glycosylated products and substrates. Michaelis-Menten plot was fitted.

### Scaled-up reactions

To prepare the glycosylated products, the reaction mixtures contained 650 µL buffer (50 mM $NaH_2PO_4$-$Na_2HPO_4$, pH 8.0), 15 µL sugar acceptor (50 mM dissolved in dimethyl sulfoxide), 100 µL NAD+ (50 mM), 20 µL UDP-GlcA (50 mM), 1.5 mg of UAXS, and 1.0 mg of GuApiGT. A total of 60 parallel tubes were used. The reactions were performed at 37 °C overnight and terminated by adding two-fold volume of methanol. The mixtures were then centrifuged at 21,130 × $g$ for 30 min. The organic solvent was removed under reduced pressure. The residue was dissolved in 1.0-1.5 mL of methanol. The products were then purified by reversed-phase semi-preparative HPLC. The structures were characterized by HRMS and extensive 1D and 2D NMR analyses. The processes for L369/H373Q and I136T/L369/H373Q mutants were similar to that of GuApiGT.

### Crystallization

The full-length cDNA of *GuApiGT* was cloned into pET-28a(+) vector. The S-tag of pET28a was removed. A TrxA-tag and 6×His-tag followed by thrombin site were added before the N-terminus of the target protein to facilitate purification. The TrxA-His-thrombin-GuApiGT protein was expressed in *E. coli* (DE3) strain and purified by Ni affinity chromatography (GE Healthcare). After purification, the recombinant protein was digested by thrombin to remove tag. The sample was mixed with Ni-NTA affinity beads for the second time to purify the protein. The flow-through was concentrated and then applied to size-exclusion chromatography on a Superdex™ 200 increase 10/300 GL prepacked column (GE Healthcare) for further purification. The elution buffer was 20 mM Tris-HCl (pH 7.5) and 50 mM NaCl. Fractions containing GuApiGT were collected and concentrated to 20 mg/mL, flash-frozen on liquid nitrogen, and then stored in a -80°C freezer. The purified protein was incubated with 5 mM UDP or UDP-Glc for 1 h. The crystals of GuApiGT were obtained after 14 days at 16 °C in hanging drops containing 1 µL of protein solution and 1 µL of reservoir solution (0.2 M lithium sulfate monohydrate, 0.1 M Bis-Tris pH 5.25, 28% *w/v* polyethylene glycol 3,350) (Supplementary Fig. 157). The crystals were flash-frozen in the reservoir solution supplemented with 25% (*v/v*) glycerol. The crystals of Sb3GT1 were obtained after 14 days at 16°C in hanging drops containing 1 µL of protein solution and 1 µL of reservoir solution (0.2 M sodium malonate pH 4.0, 20% *w/v* polyethylene glycol 3,350). The crystals of Sb3GT1-375S/Q377H were obtained after 14 days at 16 °C in hanging drops containing 1 µL of protein solution and 1 µL of reservoir solution (0.05 M citric acid, 0.05 M Bis-Tris propane pH 5.0, 16% *w/v* polyethylene glycol 3,350).

### Crystal structure determination

The diffraction data of GuApiGT and Sb3GT1 crystals were collected at beamlines BL19U1 and BL02U1 Shanghai Synchrotron Radiation Facility (SSRF). The data were processed with XDS. The structures were solved by molecular replacement with Phaser. Crystallographic refinement was performed repeatedly using Phenix and COOT. The refined structures were validated by Phenix and the PDB validation

server (https://validate-rcsb-1.wwpdb.org/). The final refined structures were deposited in the Protein Data Bank. The diffraction data and structure refinement statistics are given in Supplementary Tables 4 and 7.

## Molecular docking

Since all the reported UGT structures are highly conserved for the UDP-sugar binding domain, we simulated the initial GuApiGT/UDP-sugar complex structures by superimposing the UDP parts of UDP-Api, UDP-Glc, and UDP-Xyl to reported structures. The binding modes of sugar acceptors to the UDP-sugar-bound GuApiGT and its mutants were derived using the Glide module[53,54] of the Schrödinger Suite (version 2021-4). The grid center for the docking of UDP-sugar was adopted to the geometrical center of His18, Asp372, and Phe195 with the grid box dimension of 25 Å. The ligands were manually prepared in Maestro interface with the atom types and bond orders correctly assigned by Ligprep module[55]. A total of 30 docking poses were generated for each system.

## Molecular dynamics (MD)

The Desmond module[56] of Schrödinger Suite (version 2021-4) was used for MD simulations of the docked complexes. The OPLS4 force field was selected for both protein and ligand atoms[57]. An orthorhombic box was added with a 10.0 Å buffering area to the protein-ligand complex and filled with ~13800 TIP3P-type[58] water molecules. The counter ions of $Na^+$ and/or $Cl^-$ were also added to neutralize the system and to mimic the physical salt concentration of 0.15 M. The simulated temperature and pressure were maintained at 300.0 K and 1.0 atm by the Nose-Hoover chain thermostat[59] and Martyna-Bobias-Klein barostat[60], respectively. The default minimization and equilibration procedures were used before the 100-ns production simulation for each protein-ligand system. The simulation interaction analysis module was used to derive statistic data of the ligand-protein interactions during the MD simulations.

## Binding free energy calculations

We used the Prime module of Schrödinger Suite (version 2021-4) to calculate the MM/GBSA[32,61] binding free energy with the continuum solvation model VSGB (variable dielectric surface generalized Born)[62]. The binding free energy of each system was averaged from 400 snapshots evenly extracted from the 100-ns trajectory.

## Well-tempered metadynamic simulations

The well-tempered metadynamic simulations[63,64] with OPLS4 force field were performed using the desmond module for representative snapshots from conventional MD (cMD) simulations, which kept the water box and counter ions. For the collective valuables, we applied the distance between the C1''' atom of sugar donor and the glycol-site of sugar acceptor (O2'') and the angle of phosphate oxygen (O3), C1''', and O2'' atoms, with the width of 0.1 Å and wall of 2.5 to 6.5 Å and width of 1° and wall to 180°, respectively. The height of external Gaussian potential, updating interval, and the bias KTemp were assigned to 0.2 kcal/mol, 1.0 ps, and 3.4 kcal/mol, respectively. With these settings, a 500-ns well-tempered metadynamic simulation was performed for each system.

## QM/MM calculations

We selected a representative snapshot from the MD simulations of WT enzyme with UDP-Api for QM/MM calculations using the ONIOM approach[65] in Gaussian 16 (rev. C.01). The snapshot extracted from MD trajectories was preprocessed using the tleap program of AMBER package (version 18.0) to generate the forcefield topology files (.prmtop)[66]. The restraint electrostatic potential (RESP) charges were applied for the ligands[67], 1 and 1a, which were derived from fitting the Gaussian calculated electrostatic charge (HF/6-31G*) using the Antechamber module of AmberTools18. The MD snapshot was energy

minimized using the *sander* program with the Amber99SB(protein)/GAFF(ligand) forcefield[68,69], followed by exporting to the VMD MolUP plugin (version 1.7.0) for the setup of ONIOM partitions[70,71]. Water molecules beyond 4 Å of protein and ions ($Na^+$ and $Cl^-$) were removed. For the QM regions, UDP-Api was truncated at the carbon atom next to the first phosphorus atom (PDB name: PA), while the sugar acceptor was truncated to the glycosidic bond. The side-chain atoms of D115, H18, H373, I136, D372 were also selected as QM regions. Linking hydrogen atoms were automatically added into the boundary between QM and MM regions where there is a breaking covalent bond. The default scaling factors for the linked bonds in Gaussian 16 were used for the MM energy calculations. All residues and water molecules within 4 Å of sugar acceptor/UDP-Api or within 6 Å of the QM region atoms were unfrozen to move during the optimization. B3LYP/6-31G(d):AMBER and (B3LYP/6-311++G(2d,2p):AMBER)=embedcharge were used for geometry optimizations and final energy calculations, respectively. The transition state geometries were initially located by flexible reactant coordinates scan and fully optimized, which were also confirmed by the unique imaginary vibrational mode connecting the bonding and de-bonding atoms.

## Hydrogen-Deuterium Exchange Mass Spectrometry (HDX-MS)

Deuterium labeling was initiated with a 20-fold dilution in $D_2O$ buffer (100 mM phosphate, pD 7.0) of WT, L369/H373Q mutant, or I136T/L369/H373Q mutant (each 1 mg/mL). After 0.083, 0.25, 1, 10, 30, 60 and 240 min of labeling, the reaction was quenched with the addition of quenching buffer (100 mM phosphate, 4 M GdHCl, 0.5 M TCEP, pH 2.0). Samples were then injected and online digested using a Waters ENZYMATE BEH pepsin column (2.1 × 30 mm, 5 μm). The peptides were trapped and desalted on a VanGuard Pre-Column trap (ACQUITY UPLC BEH C18, 1.7 μm), eluted with 15% aqueous acetonitrile at 100 μL/min, and then separated on an ACQUITY UPLC BEH C18 column (1.7 μm, 1.0 × 100 mm). All mass spectra were acquired on a Waters Xevo G2 mass spectrometer, and processed using DynamX 3.0 software. Peptides from an unlabeled protein were identified using ProteinLynx Global Server (PLGS) searches of a protein database including WT, L369/H373Q mutant, and I136T/L369/H373Q sequences. Relative deuterium levels for each peptide were calculated by subtracting the mass of the undeuterated control sample from that of the deuterium-labeled sample. Deuterium levels were not corrected for back exchange and were thus reported as relative[35].

## De novo biosynthesis of flavonoid apiosides in tabacco

The full-length DNA regions of *AtPAL*, *AtC4H*, *At4CL*, *AtCHS*, *GuCHR*, *GuCHI*, *AtCHI*, *PcFNSI*, *pgm*, *GalU*, *CalS8*, *UAXS*, *GuGT14*, *GuGT53*, and *GuApiGT* were amplified using primers given in Supplementary Data 1. The PCR products were subcloned into pDonr207 vectors using the Gateway BP Clonase II Enzyme Mix and then cloned into pEAQ-HT-DEST1 vector using the Gateway LR Clonase II Enzyme Mix according to the manufacturer's instructions. The recombinant pEAQ-HT-DEST1-*GuApiGT* vector was transformed into *Agrobacterium tumefaciens* strain GV2260 by chemical conversion method.

Single colonies were inoculated at 28 °C with shaking in LB culture medium (50 μg/mL kanamycin and 50 μg/mL rifampicin) until $OD_{600}$ = 0.6. After centrifugation, bacteria were re-suspended in MMA buffer to $OD_{600}$ = 0.2 for each strain. Different strains were mixed for transformation. The infection solution was infiltrated into leaves of 5-6 week-old tobacco. After 7 days, the samples were harvested and freeze-dried. The secondary metabolites were extracted by 50% (v/v) methanol and analyzed by LC/MS.

The contents of **1a** and **2a** were quantified by regression equations, which were also used for semi-quantification of the other 8 flavonoid apiosides. Reference standards **1a** and **2a** were respectively dissolved in DMSO to make solutions of 2 mg/mL and 1 mg/mL, which were 1:1 mixed to obtain the mixed stock solution. The stock solution

was serially diluted using 50% methanol to obtain calibration standard solutions (diluted by 2, 4, 8, 16, 32, 64, 128 and 256 folds, respectively). The regression equations of **1a** and **2a** were $y = 1.2105e^5x + 7.1413e^6$ ($r^2 = 0.999$), and $y = 7.713e^4x + 3.388e^6$ ($r^2 = 0.998$), respectively, where $x$ represents the concentration (ng/mL), $y$ the peak area, and $r$ the correlation coefficient.

## Reporting summary

Further information on research design is available in the Nature Portfolio Reporting Summary linked to this article.

## Data availability

Data supporting the findings of this study are available in the article, supplementary materials, or public database. The gene sequence data generated in this study have been deposited in the NCBI database under the following accession numbers: GuApiGT (OQ201607), SsApiGT/PtApiGT/GiApiGT/GgApiGT (OQ230794-OQ230797), GuGT53 (OQ266890), and other apiosyltransferase candidate genes from Leguminosae plants (OR372660-OR372775). The raw reads from the RNA-sequencing profiling analysis of *Glycyrrhiza uralensis* have been deposited in the NCBI Sequence Read Archive (SRA) database under the BioProject accessions PRJNA945816. The crystal structures in this study have been deposited in the RCSB PDB database under the following accession numbers: GuApiGT (8HZZ), Sb3GT1 in complex with UDP (8IOE), and Sb3GT1-375S/Q377H in complex with UDP-Glc (8IOD). The primers and Gaussian optimized geometries (RC, TS, and PC) are given in Supplementary Data 1–4. Source data are provided with this paper.

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

## Acknowledgements

This work was supported by National Natural Science Foundation of China (Grants No. 81891010/81891011, 82330122 and 81725023 to M.Y.; 82122073 to X.Q.), China National Postdoctoral Program for Innovation Talents (Grant No. BX20220022 to Z.L.W.), and China Postdoctoral Science Foundation (Grant No. 2023M730131 to Z.L.W.). We thank Dr. Xiao-Meng Shi and Dr. Hong-Li Jia at State Key Laboratory of Natural and Biomimetic Drugs of Peking University for assistance in HDX-MS and X-ray diffraction experiments. We thank Professor Qing Jin at Anhui Agricultural University for assistance in plant materials collection. We thank Prof. George Lomonossoff at John Innes Centre for providing the pEAQ-HT vector. We thank the staff at BL19U1/BL02U1 beamlines at SSRF of the National Facility for Protein Science in Shanghai (NFPS), Shanghai Advanced Research Institute, Chinese Academy of Sciences, for providing technical support in X-ray diffraction data collection and analysis. The computations were enabled by resources provided by the Swedish National Infrastructure for Computing (SNIC) at the National Supercomputer Center (Grant No. SNIC2022-3-34) at Linköping University (Sweden).

## Author contributions

M.Y., Z.L.W. and X.Q. designed research and acquired funding. F.D.L. supervised the crystallography experiments. J.H.L. and H.Å. contributed the theoretical calculation. H.T.W. and Z.L.W. designed and performed

all experiments and analyzed the data. K.C., M.J.Y., M.Z., R.S.W. and J.H.Z. assisted with experiments; H.T.W., Z.L.W., J.H.L. and M.Y. wrote the manuscript. All authors have given approval to the final version of the manuscript.

## Competing interests

The authors declare no competing interests.
