## [Peer Review File · Nature Communications]

REVIEWER COMMENTS

Reviewer #1 (Remarks to the Author):

Wang et al discovered the first apiosyltransferase, GuApiGT, from the medical plant *Glycyrrhiza uralensis* and characterized its catalytic mechanism through crystallography, mutagenesis, phylogenetic analysis, and plentiful calculations. The conclusions are convincing and were further verified by applied mutagenesis in GuApiGT and heterologous glycosyltransferases. Based on the newly gained apiotransferases and knowledges, the authors built up de novo biosynthetic pathway of several bioactive apiosides in *Nicotiana benthamiana*. The findings are novel and practical in the biosynthesis of pharmaceutically important apiosides.

Here are some small questions:

- 1) Line 281-283, "The peptide L135-E156 also changed remarkably, probably due to the hydrogen bond interaction with glucose 6-OH after I136T mutation" It was not clearly explained here. Did the authors mean the increasement in molecular weight derived from the hydrogen-deuterium exchange between 136T and glucose 6-OH? This hypothesis looks unreasonable.
- 2) Line 304-307, the authors stated that the declined catalytic efficiency of mutants come from the increased flexibility of the NTD, which has low affinity to glycoside. Actually, L369/H373Q, G370/H373Q and I136T/G370/H373Q mutants displayed comparable catalytic efficiency to WT, suggesting the mutagenesis impair their affinity to UDP-sugar rather than the substrate binding capacity. Please give more reasonable explain to the results. Besides, "substrate" in this paragraph should be "glycoside" or "sugar acceptor".
- 3) Line 318-338, although the Sb3GT1 375S/Q377H mutant obtained larger UDP-sugar binding motif, it can only recognize UDP-Api in a very low efficiency. Is there some explain for that? The authors concluded that R368 of GuApiGT is key residue in the recognition of UDP-Api, why didn't introduce F372R mutation into Sb3GT1 to further verify this hypothesis?
- 4) Line 523-524, "The mobile phase was a gradient elution of solvents A (acetonitrile, ACN) and B (water containing 0.1% formic acid)", the solvents are different from that in Supplementary Table 4.

Reviewer #2 (Remarks to the Author):

The manuscript entitled 'Discovery, mechanisms, and engineering of the missing apiosylation step in the biosynthesis of flavonoid aposides in Leguminosae plants' reported an apiosyltransferase GuApiGT from medicinal plant *G. uralensis*. Biochemical characterization supported that GuApiGT specifically utilizes UDP-apiose as a sugar donor and catalyzes 2''-O-apiosylation of flavonoid glycosides. The binding site for UDP-apiose was determined through structural determination, theoretical calculations, and metagenesis analysis. This enzyme also shows broad substrate promiscuity, it accepts 37 glycosides of flavonoids, liganans, and coumarins. Several new compounds were generated and may show bioactivity. The authors successfully performed de novo biosynthesis to produce various flavonoid apiosides in tobacco. This work discovered 121 candidate apiosyltransferase genes from Leguminosae plants and determined the first crystal structure of apiosyltransferase. The key apiose binding motif (RLGSDH) was identified, providing important information for engineering of glycosyltransferases. The results possess significant novelty, the presentation is readable. Below are some of my major and minor concerns regarding the article:

Major Concern:

1. The authors mention many times that 'no reported UGTs could accept UDP-apiose as the sugar donor'; however, there is a research article, Identification of an apiosyltransferase in the plant pathogen *Xanthomonas pisi* (PLoS ONE 13(10) 2018), reported the first evidence of an apiosyltransferase (XpApiT). Although the sugar acceptor of XpApiT was unknown, indirect activity assay (UDP-Glo) and in microbe co-expression supported that XpApiT specifically utilizes UDP-apiose as sugar donor. Does XpApiT possess the conserved apiose-binding residues or motifs, such as RLGSDH motif? Please rephrase the relevant sentences, 'no reported.....', into more appropriate ones.
2. GuApiGT catalyzes the formation of glycosidic bond between O-2'' of the glucose residue (Compound 2) and C-1''' of the Api moiety (UDP-Api). However, in the docking model shown in Fig. 4e, O-2'' of the glucose residue seems to be far away from C-1''' of the Api moiety. Is this a

reasonable distance for catalysis? Please label the distance between O-2'' of the glucose residue (Compound 2) and C-1''' of the Api moiety (UDP-Api).

3. In Fig. 4g, the authors show QM/MM optimized geometry of transition state (TS), where O-2'' of the glucose residue is covalent bonded with H18. In general, D115 facilitates H18 to deprotonate the hydroxyl group of O-2''. The covalent bond between O-2'' and C-1''' should not be the TS.

Furthermore, the authors did not describe the roles of H18 and D115. Are these two residues conserved among the GuApiGT homologues? Please regenerate Fig. g and describe the function of H18 and D115.

Minor Concern:

1. Page 3: Line 51, 'form' should be 'from'.

2. Page 11: Line 208, the authors should tell readers what and where the PSPG box is.

3. In Fig. 4f, the authors only show a sugar donor, UDP-Api. The figure legend does not match to the figure.

4. Fig. 4g is not very clear. The structure of Api appears to be incorrect. Please generate a new one to make it more accessible.

5. GuApiGT catalyzes the formation of glycosidic bond between O-2'' of the glucose residue (Compound 2) and C-1''' of the Api moiety (UDP-Api). The labeling of C1' and O5 in Fig. f is confusing. Please use C1''' and O2'' instead of C1' and O5 in Fig. f and the text.

6. The 'Number of atoms' of Sb3GT1 375S/Q377H in Table 8 does not match to the crystal structure. Please revise Table 8.

Reviewer #3 (Remarks to the Author):

Recommendation: Accept after major revisions.

The paper presented by Ye and colleagues delivers a comprehensive investigation of a previously unreported apiosyltransferase, GuApiGT, from *Glycyrrhiza uralensis*. Their study elegantly elucidates the mechanism of GuApiGT's role in catalyzing the 2''-O-apiosylation of flavonoid glycosides. This has been achieved by analyzing the enzyme's structure alongside related Sb3GT1 crystal structures, and supported by theoretical calculations. Furthermore, they have successfully generated mutants with altered sugar selectivity via protein engineering, thereby expanding the boundaries of understanding of this novel enzyme family. The unique apiosyltransferase discovery and subsequent characterization are well executed, notably identifying the critical RLGSDH motif involved in substrate recognition. The authors have also pioneered a de novo biosynthesis in *Nicotiana benthamiana* to produce liquiritin apioside, a significant advancement in this field. Overall, this paper is well-constructed, clearly laying out the logic underpinning the enzyme characterization, supported by compelling results and robust data.

However, it is important to note that a preprint paper titled "Discovery of the apiosyltransferase, celery UGT94AX1 that catalyzes the biosynthesis of a flavone glycoside, apiin" was posted in bioRxiv very recently. This paper similarly claims the identification of a plant apiosyltransferase, in this case, celery UGT94AX1 (AgApiT), and there seems to be a significant overlap in the content of the two papers. Thus, it would be beneficial to revise sections such as lines 74-76, 360-363, and 435-436 to emphasize the unique features of GuApiGT in comparison to AgApiT. Highlighting these distinctions will reinforce the novelty of your study.

Nevertheless, the discovery of GuApiGT remains intriguing and constructive. There are, however, several points that I believe warrant attention and further clarification:

Major Revisions:

1. It seems unusual that enzyme kinetic studies are not part of your enzyme characterization. For novel enzymes such as GuApiGT, key kinetic parameters, like K_M , V_{max} , and k_{cat} , are essential to understand its behavior. Such studies could yield valuable insights into the enzyme's reaction efficiency and substrate affinity. I would strongly recommend adding an enzyme kinetics study to strengthen the thoroughness of your characterization.

2. It would be beneficial to provide context for the selection of the 15 reported UGTs in Supplementary Fig.1. If there's a specific reason for this choice, or it's based on a review or database, providing this information would aid comprehension. Additionally, the significance of the purple font is unclear.
3. In Fig.2a, it appears the LC peak may be truncated or edited. Please revisit your original data to verify the accuracy of this representation.
4. Unlike the engineered Sb3GT1, GuApiGT accepts flavonoid, lignan, or coumarin glycosides, but not free aglycones. An explanation based on structural evidence would be enlightening. Moreover, an explanation for the regioselectivity of GuApiGT towards 5-O-glycosides instead of 3-O-glycosides, as informed by the solved structures, would be informative.
5. While obtaining crystal structures is commendable, a comparison of GuApiGT with other GTs to understand why they cannot perform apiosyltransferase functions would be very insightful.
6. Performing enzymatic activity assays with the mutant enzyme in the presence of UDP-Xyl and UDP-Glc would offer direct evidence of substrate turnover, greatly reinforcing your claim of substrate binding.
7. The significant $\Delta\Delta E$ for GLU272 warrants its inclusion in mutagenesis studies. Also, the significance of the bold font in Supplementary Table 7 should be clarified. Lastly, consistency in amino acid notation between the main text and supplementary information is necessary.

Additional Minor Revisions:

1. This sentence needs revision: "In this work, we report the first apiosyltransferase GuApiGT, which from medicinal plant *Glycyrrhiza uralensis*." A potential revision could be: "In this work, we report the discovery of GuApiGT, the first apiosyltransferase derived from the medicinal plant *Glycyrrhiza uralensis*."
2. Several lines seem to lack necessary citations. Please add the appropriate references to lines 42, 50, 52, 71, and 370.
3. In lines 57-62, the identification of *Glycyrrhiza uralensis* Fisch. as the target plant could be more solidly substantiated, considering the existence of multiple popular medicinal plants within the Leguminosae family. Also, the discussion about liquiritin apioside could be made more relatable to lines 84-86.
4. In Figure 4d, the electron density map appears convoluted, which may confuse the readers. Consider refining it for enhanced clarity.
5. In Figures 4e and 4g, the rationale behind presenting some residues as lines and some as sticks is not evident. Furthermore, the coloring of R368 in orange is not explained in the figure legend.
6. The term "directed evolution" might not accurately describe your engineering efforts. Perhaps "rational design" or "protein engineering" would be more fitting.
7. A clearer statement of the structural superimposition could be "The R368L369G370S371D372H373 motif in GuApiGT is mapped to F372F373G374D375Q376 of Sb3GT1."
8. The error bar in Figure 5c appears inconsistent with others. Please review and rectify if necessary.
9. The notation "375S" could potentially confuse the readers. I recommend providing further clarification in line 327, 330, Figure 6c, and Supplementary Figure 148.
10. The differences in visualization methods used in Fig. 6a and 6c (sticks vs. lines) and the presence of a white cartoon in the zoomed image of Fig. 6a should be adequately justified in the figure legends.
11. The quench procedure is described differently in line 520 ("ice-cold methanol") and line 545 ("pre-cooled methanol (MeOH)"). Please ensure consistency. Additionally, providing an abbreviation for

methanol (MeOH) in line 545 seems redundant, as it was not used subsequently.

12. In line 552, the statement would be clearer as "two-fold volume of methanol."

13. For Extended Data Figure 1, specifying the contour level and carve radius used for the electron density maps would greatly improve clarity and interpretability. This information would aid the readers in assessing the quality of the depicted substrate-protein interactions.

14. In Supplementary Fig.2, at least three impurities are present in lane 1, which contradicts the claim of achieving 95% purity.

15. Despite the ambiguity surrounding the effect of Ba^{2+} and Cu^{2+} on GuApiGT activity, it is inaccurate to describe the enzyme as divalent ion-independent. Please also ensure charges are correctly superscripted in figures.

16. Clarification of the peak proximal to 2a in Supplementary Fig. 137 is necessary. Is it a product of GuApiGT or something else?

Responses to Reviewers' comments

Reviewer #1:

Wang et al discovered the first apiosyltransferase, GuApiGT, from the medical plant *Glycyrrhiza uralensis* and characterized its catalytic mechanism through crystallography, mutagenesis, phylogenetic analysis, and plentiful calculations. The conclusions are convincing and were further verified by applied mutagenesis in GuApiGT and heterologous glycosyltransferases. Based on the newly gained apiotransferases and knowledges, the authors built up *de novo* biosynthetic pathway of several bioactive apiosides in *Nicotiana benthamiana*. The findings are novel and practical in the biosynthesis of pharmaceutically important apiosides.

R: We thank the reviewer for carefully reading our manuscript, and giving us the valuable suggestions and comments.

Here are some small questions:

1) Line 281-283, “The peptide L135-E156 also changed remarkably, probably due to the hydrogen bond interaction with glucose 6-OH after I136T mutation” It was not clearly explained here. Did the authors mean the increasement in molecular weight derived from the hydrogen-deuterium exchange between 136T and glucose 6-OH? This hypothesis looks unreasonable.

R: The increasement in molecular weight is not derived from the exchange between T136 and glucose 6-OH. Instead, it is derived from the hydrogen-deuterium exchange between the protein (hydrogen atoms at the outer surface of the protein) and the solvent D₂O. After mutagenesis, the increasement in molecular weight of certain peptides may be different for the wild type and the mutant, as mutagenesis may change conformation of the protein. The hydrogen-deuterium exchange mass spectrometry (HDX-MS) technique was firstly published in 1993 (*J. Am. Chem. Soc.* 1993, 115, 6317-6321; *Protein Sci.* 1993, 2, 522-531). Currently, it has been developed into a popular tool to dissect the catalytic mechanisms of enzymes (*Chem. Rev.* 2022, 122, 7562-7623) (Fig. R1). In the present work, the molecular weight of peptide L135-E156 increased by

around 2 Da when I136 was mutated to threonine (Fig. 5e). This result demonstrated the significance of T136 in recognizing UDP-Glc. To avoid any ambiguity, we have revised this sentence into “The L135-E156 peptide of I136T/L369/H373Q mutant exhibited remarkable increase of deuterium uptake, verifying the significance of T136 in recognizing UDP-Glc (Supplementary Figures137).”.

Fig. R1 A general mechanism for hydrogen-deuterium exchange mass (HDX-MS). (Adapted from *Chem. Rev.* 2022, 122, 7562-7623)

Fig. 5e Deuterium uptake plots of peptide 135-156 at different time points.

2) Line 304-307, the authors stated that the declined catalytic efficiency of mutants come from the increased flexibility of the NTD, which has low affinity to glycoside. Actually, L369/H373Q, G370/H373Q and I136T/G370/H373Q mutants displayed comparable catalytic efficiency to WT, suggesting the mutagenesis impair their affinity to UDP-sugar rather than the substrate binding capacity. Please give more reasonable explain to the results. Besides, “substrate” in this paragraph should be “glycoside” or “sugar acceptor”.

R: Actually, the catalytic efficiency of mutants was decreased when compared with the native. Conversion rates of the mutants could reach high levels only by adding 8 folds of enzyme (40 μ g mutant enzyme, UDP-Xyl or UDP-Glc as sugar donor; 5 μ g native enzyme, UDP-Api as sugar donor). The native enzyme showed higher activities than the mutants under the same reaction conditions (Fig. R2).

Generally, the C-terminal domain (CTD) and N-terminal domain (NTD) of UGTs are responsible for sugar donor binding and sugar acceptor binding, respectively (*Biotechnol. Adv.* 2022, 60, 108030). In the present work, HDX-MS analysis indicated increased flexibility of the NTD in the mutant enzymes, which may cause decreased affinity between the sugar acceptor and the enzyme. These deductions were consistent with enzyme assay results.

In addition, we have revised “substrate binding” to “sugar acceptor” in this manuscript.

Fig. R2 The glycosylation conversion rates of the wild type (WT), L369/H373Q, G370/H373Q, I136T/G370/H373Q mutants, using **2** as sugar acceptor.

3) Line 318-338, although the Sb3GT1 375S/Q377H mutant obtained larger UDP-sugar binding motif, it can only recognize UDP-Api in a very low efficiency. Is there some explain for that? The authors concluded that R368 of GuApiGT is key residue in the

recognition of UDP-Api, why didn't introduce F372R mutation into Sb3GT1 to further verify this hypothesis?

R: For wild type Sb3GT1, F373 has π - π interactions with kaempferol (Fig. R3). While the 375S/Q377H mutant could recognize UDP-Api, the introduction of S375 led to the loss of interactions between the binding pocket and the sugar acceptor. Thus, the apiosylation activity of the mutant was low.

R368 is indeed a key residue in recognizing UDP-Api. According to the reviewer's suggestion, we constructed F372R/Q376H and F372R/375S/Q377H mutants. As expected, the F372R/Q376H mutant showed noticeable apiosylation activity (Fig. R4). These results have been included into Figure 6b.

Fig. R3 The key binding motif in Sb3GT1/UDP and Sb3GT1 375S/Q377H mutant/UDP-Glc. The key amino acids were labeled by red color font. Kaempferol was docked into Sb3GT1 and its mutant based on the crystal structure of VvGT1 (PDB ID: 2C1Z). The hydrogen-bond interaction and π - π interaction are shown as yellow and purple dashes, respectively.

Fig. R4 Functional characterization of Sb3GT1 and its mutants. Control 1, the acceptor was incubated with Sb3GT1 and an UDP-Api supply system (UDP-GlcA, UAXS, NAD⁺). Control 2, the acceptor was incubated with Sb3GT1 and UDP-Xyl. Proposed structures for **66a**, kaempferol 3-*O*-apioside; **66b**, kaempferol 3-*O*-xyloside.

4) Line 523-524, “The mobile phase was a gradient elution of solvents A (acetonitrile, ACN) and B (water containing 0.1% formic acid)”, the solvents are different from that in Supplementary Table 4.

R: We are sorry for this mistake. The description in the Supplementary Table 4 is correct. We have corrected this mistake.

Reviewer #2:

The manuscript entitled ‘Discovery, mechanisms, and engineering of the missing apiosylation step in the biosynthesis of flavonoid apiosides in Leguminosae plants’ reported an apiosyltransferase GuApiGT from medicinal plant *G. uralensis*. Biochemical characterization supported that GuApiGT specifically utilizes UDP-apiose as a sugar donor and catalyzes 2''-*O*-apiosylation of flavonoid glycosides. The binding site for UDP-apiose was determined through structural determination,

theoretical calculations, and metagenesis analysis. This enzyme also shows broad substrate promiscuity, it accepts 37 glycosides of flavonoids, lignans, and coumarins. Several new compounds were generated and may show bioactivity. The authors successfully performed de novo biosynthesis to produce various flavonoid apiosides in tobacco. This work discovered 121 candidate apiosyltransferase genes from Leguminosae plants and determined the first crystal structure of apiosyltransferase. The key apiose binding motif (RLGSDH) was identified, providing important information for engineering of glycosyltransferases. The results possess significant novelty, the presentation is readable. Below are some of my major and minor concerns regarding the article:

R: We thank the reviewer for carefully reading our manuscript, and giving us the valuable suggestions and comments.

Major Concern:

1. The authors mention many times that ‘no reported UGTs could accept UDP-apiose as the sugar donor’; however, there is a research article, Identification of an apiosyltransferase in the plant pathogen *Xanthomonas pisi* (PLoS ONE 13(10) 2018), reported the first evidence of an apiosyltransferase (XpApiT). Although the sugar acceptor of XpApiT was unknown, indirect activity assay (UDP-Glo) and in microbe co-expression supported that XpApiT specifically utilizes UDP-apiose as sugar donor. Does XpApiT possess the conserved apiose-binding residues or motifs, such as RLGSDH motif? Please rephrase the relevant sentences, ‘no reported.....’, into more appropriate ones.

R: Thank you very much for this constructive comment. We did have noticed XpApiT (*PLoS One*, 2018, 13, e0206187). While it may be a potential apiosyltransferase, the authors did not characterize structure of the enzyme catalysis product.

Following the reviewer’s suggestion, we analyzed the sequence information of XpApiT, but did not find the RLGSDH motif characteristic for Leguminosae ApiGTs discovered in our study. In fact, we did not find the conserved PSPG box for UGTs (Fig. R5).

Very recently, Anne Osbourn's group from John Innes Centre reported the first characterized apiosyltransferase UGT73CY2 (*Science*, 2023, 379, 1252-1264. Mar 23, 2023). This enzyme could catalyze the apiosylation of one triterpenoid saponin. Besides this, very limited information on the catalytic feature of UGT73CY2 was provided in this paper. It is not known whether UGT73CY2 could catalyze other types of substrates, or whether it specifically accepts UDP-Api. Its sequence contains the PSPG box, but not the RLGSDH motif. Thus, GuApiGT and the other four Leguminosae ApiGTs represent a novel group of apiosyltransferase.

We want to mention that the *Science* paper was published after the initial submission of our paper (Mar 13, 2023, NATCATAL-23030347). Therefore, we did not read this paper when we were preparing our manuscript.

Based on the above situations, we have revised our statements in the manuscript into "To our best knowledge, GuApiGT is the first phenolic apiosyltransferase that has been reported." As no evidences prove that UGT73CY2 could accept phenolic compounds as substrate, it should be reasonable to state that GuApiGT is the first apiosyltransferase that can utilize phenolic compounds.

Fig. R5 Amino acid sequence comparison of XpApiT, UGT73CY2, GuApiGT, GgApiGT, GiApiGT, PtApiGT, and SsApiGT. PSPG box is marked in blue, and the RLGSDH motif is marked in green. This figure was produced with ENDscript (<http://multalin.toulouse.inra.fr/multalin/>).

2. GuApiGT catalyzes the formation of glycosidic bond between O-2'' of the glucose residue (Compound 2) and C-1''' of the Api moiety (UDP-Api). However, in the docking model shown in Fig. 4e, O-2'' of the glucose residue seems to be far away from

C-1''' of the Api moiety. Is this a reasonable distance for catalysis? Please label the distance between O-2'' of the glucose residue (Compound 2) and C-1''' of the Api moiety (UDP-Api).

R: The distance between O-2'' and C-1''' is 3.4 Å. We have added this number into Figure 4e. Moreover, we analyzed the reported ternary complex structures of glycosyltransferases. The distance between the catalytic site of sugar acceptor and C-1 of UDP-sugar is 4.3 Å in 2C1Z (VvGT1) and 3.8 Å in 2VCE (UGT72B1), respectively (Fig. R6). Thus, the distance of 3.4 Å for GuApiGT is reasonable for the glycosylation catalytic reaction to occur.

Fig. 4e A representative configuration of GuApiGT/UDP-Api/2 extracted from MD simulations. The hydrogen-bond interactions and π - π /cation- π interactions are shown as yellow and purple dashes, respectively. The key amino acids interacted with ligands are highlighted using sticks. The unique R368 is depicted as orange sticks, the others as blue. The other amino acids in key motif are depicted using lines.

Fig. R6 Structural analysis of 2C1Z and 2VCE.

3. In Fig. 4g, the authors show QM/MM optimized geometry of transition state (TS), where O-2'' of the glucose residue is covalent bonded with H18. In general, D115 facilitates H18 to deprotonate the hydroxyl group of O-2''. The covalent bond between O-2'' and C-1''' should not be the TS. Furthermore, the authors did not describe the roles of H18 and D115. Are these two residues conserved among the GuApiGT homologues? Please regenerate Fig. g and describe the function of H18 and D115.

R: Thank you for this comment. In the transition state, O-2'' and C-1''' should not form a covalent bond. Actually, the yellow lines in the previous version of Fig. 4g did not mean covalent bonds. We are sorry for this misleading. During this revision, we have revised Fig. 4g, where key distances and angles are shown in magenta dashes.

We have also compared the sequences of GuApiGT and the other ApiGTs. H18 and D115 are highly conserved (Fig. R7). We have added descriptions on their roles in the manuscript: “During the process to form the glycosidic bond between O2'' of **2** and C1''' of UDP-Api, H18 could partially deprotonate **2**, with the assistance of D115. Once the reaction is completed, D115 is protonated in the product complex.”.

Fig. 4g QM/MM optimized geometry of transition state (TS) at the theory of B3LYP/6-311++g(2d,2p): amber with the electronic embedding scheme and thermal zero-point energy calculated from the theory of B3LYP/6-31g(d): amber. The QM region atoms, hydrogen bonds, and key angle and distances are highlighted in green sticks, yellow dashes, and magenta dashes, respectively. The MM region atoms are depicted using

lines.

Fig. R7 Sequence alignment of GuApiGT and 4 other ApiGTs. H18 and D115 are marked in blue. This figure was produced with ENDscript (<http://multalin.toulouse.inra.fr/multalin/>).

Minor Concern:

1. Page 3: Line 51, ‘form’ should be ‘from’.

R: We have revised ‘form’ into ‘from’.

2. Page11: Line208, the authors should tell readers what and where the PSPG box is.

R: Generally, the plant secondary product glycosyltransferase box (PSPG box) contains 44 amino acids in the C-terminal domain (CTD) of UGT. It usually starts with W and ends with DQ or EQ, and contains the conserved sequence of HCGXXS, which plays an important role in sugar donor recognition. The PSPG box of GuApiGT and other ApiGTs is shown in **Supplementary Figure 143**. During this revision, we have added **Supplementary Figure 83**, which shows the position of PSPG box in the crystal structure of GuApiGT.

Supplementary Fig. 83 The PSPG box (W329-H373, colored in gray) in crystal structure of GuApiGT.

3. In Fig. 4f, the authors only show a sugar donor, UDP-Api. The figure legend does not match to the figure.

R: Actually, Fig. 4f contains four pictures. The metadynamic simulations for UDP-Api, UDP-Glc, and UDP-Xyl are all shown in this figure. During this revision, we have added a frame box in Fig. 4f for easy understanding.

Fig. 4f Metadynamic simulations of GuApiGT with different sugar donors (Api, Xyl, and Glc). CV1, the distance of O2''-C1''' (Å); CV2, the angle of O2''-C1'''-O3 (°).

4. Fig. 4g is not very clear. The structure of Api appears to be incorrect. Please generate a new one to make it more accessible.

R: The structure of Api should be correct. For easy reading, we have adjusted the figure to clearly demonstrate the structure of Api (Fig. 4g).

5. GuApiGT catalyzes the formation of glycosidic bond between O-2'' of the glucose residue (Compound 2) and C-1''' of the Api moiety (UDP-Api). The labeling of C1' and O5 in Fig. f is confusing. Please use C1''' and O2'' instead of C1' and O5 in Fig. f and the text.

R: We had used the PDB atom labelling (C1' and O5) in the previous version. To avoid any confusion, we used C1''' and O2'' in the revised manuscript (Fig. 4f).

6. The 'Number of atoms' of Sb3GT1 375S/Q377H in Table 8 does not match to the crystal structure. Please revise Table 8.

R: We are sorry for this mistake. These data have been corrected in Table 8.

Table 8. Data collection and refinement statistics of Sb3GT1 crystals.

	Sb3GT1/UDP	Sb3GT1 375S/Q377H complex with UDP-Glc
Wavelength(Å)	0.97918	0.97918
Space group	P 21 21 2	P 21 21 21
Cell parameters		
a, b, c (Å)	101.65, 61.28, 68.64	47.47, 74.52, 128.98
α , β , γ (°)	90, 90, 90	90, 90, 90
Resolution(Å)	45.71-1.9(1.94-1.90) ^a	47.47-1.43(1.45-1.43) ^a
R_{merge} (%)	10.6 (133)	4.3(103)
$CC_{1/2}$ (%)	99.9(75.3)	100(78)
$I/\sigma I$	18.6(2.2)	24.8(2.1)
Completeness (%)	99.0 (96.8)	100(100)
Average redundancy	13 (10.8)	12.8(10.5)
Refinement		
No. reflections (overall)	34678	85370
No. reflections (test set)	1706	4352
$R_{\text{work}}/R_{\text{free}}$ (%)	19.99/23.89	14.59/18.43
Number of atoms		
Protein	3425	3514

H ₂ O	96	222
SO ₄		
UDP	25	
UDPGlc		36
B factors (Å ²)		
Protein	35.81	29.92
H ₂ O	35.02	36.37
SO ₄		
UDP	36.64	
UDPGlc		34.33
r.m.s. deviations		
Bond lengths (Å)	0.0063	0.0155
Bond angles (°)	1.4130	1.9494
Ramapage plot % residues		
Favored	97.21	97.49
Allowed	2.79	2.28
Outliers	0	0.23

^a Values in parentheses are for highest-resolution shell.

Reviewer #3:

The paper presented by Ye and colleagues delivers a comprehensive investigation of a previously unreported apiosyltransferase, GuApiGT, from *Glycyrrhiza uralensis*. Their study elegantly elucidates the mechanism of GuApiGT's role in catalyzing the 2''-*O*-apiosylation of flavonoid glycosides. This has been achieved by analyzing the enzyme's structure alongside related Sb3GT1 crystal structures, and supported by theoretical calculations. Furthermore, they have successfully generated mutants with altered sugar selectivity via protein engineering, thereby expanding the boundaries of understanding of this novel enzyme family. The unique apiosyltransferase discovery and subsequent characterization are well executed, notably identifying the critical RLGSDH motif involved in substrate recognition. The authors have also pioneered a *de novo* biosynthesis in *Nicotiana benthamiana* to produce liquiritin apioside, a significant advancement in this field. Overall, this paper is well-constructed, clearly laying out the logic underpinning the enzyme characterization, supported by compelling results and robust data.

However, it is important to note that a preprint paper titled "Discovery of the apiosyltransferase, celery UGT94AX1 that catalyzes the biosynthesis of a flavone glycoside, apiin" was posted in bioRxiv very recently. This paper similarly claims the identification of a plant apiosyltransferase, in this case, celery UGT94AX1 (AgApiT), and there seems to be a significant overlap in the content of the two papers. Thus, it would be beneficial to revise sections such as lines 74-76, 360-363, and 435-436 to emphasize the unique features of GuApiGT in comparison to AgApiT. Highlighting these distinctions will reinforce the novelty of your study.

Nevertheless, the discovery of GuApiGT remains intriguing and constructive. There are, however, several points that I believe warrant attention and further clarification:

R: We thank the reviewer for carefully reading our manuscript, and giving us the valuable suggestions and comments. We have also noticed the preprint paper about UGT94AX1 from celery (<https://doi.org/10.1101/2023.05.22.541790>, Posted May 23, 2023). Our manuscript was posted on the preprint website Research Square on March 29, 2023. Because our manuscript was open to the public two months earlier than the UGT94AX1 paper, we believe that GuApiGT is the first phenolic apiosyltransferase that has been characterized.

It is interesting to discover UGT94AX1 as an apiosyltransferase from celery (*Apium graveolens*) of the Apiaceae family. We have compared the catalytic features of UGT94AX1 and GuApiGT. They show remarkably different features. First, UGT94AX1 shows low catalytic activity, with K_m (affinity) and k_{cat}/K_m (catalytic efficiency) values of $15 \mu\text{mol}\cdot\text{L}^{-1}$ and $5.8\cdot 10^{-5} \text{ s}^{-1}\cdot\mu\text{mol}^{-1}\cdot\text{L}$, respectively. (Fig. R8). In contrast, these values for GuApiGT were $2.59 \mu\text{mol}\cdot\text{L}^{-1}$ and $4.2\cdot 10^{-2} \text{ s}^{-1}\cdot\mu\text{mol}^{-1}\cdot\text{L}$. The catalytic efficiency (k_{cat}/K_m) was around 1000-fold different.

Second, the sequence of UGT94AX1 has a 44-amino acid PSPG box, and it does not contain the RLGSDH motif (Fig. R8b). Therefore, these features may be unique for Leguminosae apiosyltransferases. GuApiGT and the other four ApiGTs represent a novel group of glycosyltransferases.

Fig. R8 a, Functional characterization of AgApiT (UGT94AX1) (Adapted from bioRxiv, doi.org/10.1101/2023.05.22.541790). **b**, Sequence alignment of GuApiGT, GgApiGT, GiApiGT, PtApiGT, SsApiGT, and UGT94AX1. The key motif is marked in green. This figure was produced with ENDscript (<http://multalin.toulouse.inra.fr/multalin/>).

Major Revisions:

1. It seems unusual that enzyme kinetic studies are not part of your enzyme characterization. For novel enzymes such as GuApiGT, key kinetic parameters, like K_m , V_{max} , and k_{cat} , are essential to understand its behavior. Such studies could yield valuable insights into the enzyme's reaction efficiency and substrate affinity. I would strongly recommend adding an enzyme kinetics study to strengthen the thoroughness of your characterization.

R: UDP-apiose is unstable and is commercially unavailable. This is the reason why we did not present the enzyme kinetic parameters of GuApiGT. In our study, we introduced the UDP-apiose/UDP-xylose synthase (UAXS) from *Arabidopsis thaliana* into the enzyme catalysis system, which could provide UDP-apiose as sugar donor.

During this revision, we managed to obtain purified UDP-apiose. UAXS could convert UDP-glucuronic acid (UDP-GlcA) into both UDP-apiose and UDP-xylose. We used the method published by Tae Fujimori et al. (*Carbohydr. Res.* 2019, 477, 20-25.) and

separated UDP-Api and UDP-Xyl by HPLC (Supplementary Fig. 156). As UDP-Api is not stable, we concentrated the HPLC eluents in a lyophilizer, and the obtained powder was stored in a -80°C freezer before use.

Once we have obtained purified UDP-Api, we were able to determine the enzyme kinetic parameters of GuApiGT. The kinetic parameters for **2** with saturated UDP-Api were measured. The apparent K_m value for **2** was $2.59 \pm 0.23 \mu\text{mol}\cdot\text{L}^{-1}$. The V_{max} value was $0.12 \pm 0.0019 \mu\text{mol}\cdot\text{min}^{-1}\cdot\text{mg}^{-1}$. The k_{cat} value was 0.11 s^{-1} . The k_{cat}/K_m value was $0.042 \text{ s}^{-1}\cdot\mu\text{mol}^{-1}\cdot\text{L}$. These data have been provided in Supplementary Fig. 5. The experimental details have been added into the “Method” section of the manuscript.

Supplementary Fig. 156 Purification of UDP-Api by HPLC. UV detection wavelength, 262 nm.

Supplementary Fig. 5 Determination of kinetic parameters for recombinant GuApiGT (n=3). The apparent K_m value was calculated from Michaelis-Menten plot with varying concentrations of compound **2** (isoliquiritin).

Method

Preparation of UDP-apiose: The reaction mixtures contained 100 μL buffer (100 mM triethylamine phosphate, pH 8.0), 0.1 mM NAD^+ , 10 mM UDP-GlcA, and 0.48 mg of

UAXS. A total of 30 parallel tubes were used. The reactions were performed at 25°C for 4h and then centrifuged at 15,000 rpm for 30 min. The products were subsequently purified by reversed-phase HPLC. HPLC was performed on an Inertsustain AQ-C18 (5 μ m, 4.6 \times 250 mm column; GL Sciences, Tokyo, Japan) at a flow rate of 1.0 mL/min. The mobile phase was a gradient elution of solvents A (100 mM *N,N*-dimethylcyclohexylamine phosphate buffer, pH 6.5) and B (30% (v/v) ACN). A gradient elution program was used: 0 min, 100% A; 13 min, 100% A; 35 min, 33% A; 39 min, 33% A; 40 min, 100% A. The eluted fractions were monitored by measuring the absorbance at 262 nm (Supplementary Fig. 156). After freeze-drying, UDP-apiose was dissolved with triethylamine phosphate for use.

Determination of kinetic parameters: 2 μ g/mL protein, 480 μ mol/L of saturated UDP-apiose, and different concentrations of compound 2 (1, 2.5, 5, 10, 30, 40, 60, 80, 150 μ mol/L) were performed in a final volume of 25 μ L with 50 mM Na₂HPO₄-NaH₂PO₄ buffer (pH 8.0). The reactions were quenched with pre-cooled methanol after incubating at 37°C for 15 min, and then centrifuged at 15,000 rpm for 15 min. The supernatants were used for HPLC analysis. All experiments were performed in triplicate. The conversion rates in percentage were calculated from HPLC peak areas of glycosylated products and substrates. Michaelis-Menten plot was fitted.

2. It would be beneficial to provide context for the selection of the 15 reported UGTs in Supplementary Fig.1. If there's a specific reason for this choice, or it's based on a review or database, providing this information would aid comprehension. Additionally, the significance of the purple font is unclear.

R: We selected 15 reported UGTs with different sugar donor selectivity for the phylogenetic analysis (Fig. R10). However, they were mainly clustered based on substrate type and glycosylation site rather than sugar donor selectivity. GuApiGT was clustered with two glycosides 2''-OGT (ZjOGT38 and TcOGT4). For better understanding, we have prepared a new phylogenetic analysis figure in the revised manuscript (Supplementary Fig.1).

Fig. R10 a, Previous phylogenetic analysis of GuApiGT (MSTRG.23171.4) with reported UGTs. **b**, Information of the 17 UGTs.

Supplementary Fig. 1 Phylogenetic analysis of GuApiGT (MSTRG.23171.4) with reported UGTs using MEGA6 software with the maximum likelihood method. The bootstrap consensus tree inferred from 1,000 replicates was taken to represent the evolutionary history of the taxa analyzed. All the GenBank accession numbers used in this study are listed in **Supplementary Table 2**.

3. In Fig.2a, it appears the LC peak may be truncated or edited. Please revisit your original data to verify the accuracy of this representation.

R: The original HPLC chromatograms were provided in Fig. R11. The peaks at around 2 min were proteins, and these peaks were not related with the catalytic function. For better understanding, we only provided the chromatograms between 4-12 min in Fig. 2a.

Fig. R11 a,b, HPLC analysis of GuApiGT catalyzed product using **1** as the substrate. **c,d**, HPLC analysis of GuApiGT catalyzed product using **2** as the substrate. UDP-Api was produced by adding UDP-GlcA, purified UAXS, and NAD⁺ to the mixed system. The chromatographic peak with a retention time of around 2 min is the protein peak.

4. Unlike the engineered Sb3GT1, GuApiGT accepts flavonoid, lignan, or coumarin glycosides, but not free aglycones. An explanation based on structural evidence would be enlightening. Moreover, an explanation for the regioselectivity of GuApiGT towards 5-*O*-glycosides instead of 3-*O*-glycosides, as informed by the solved structures, would be informative.

R: We simulated the binding of **2** (isoliquiritin, isoliquiritigenin 4-*O*-glucoside), **2'** (isoliquiritigenin), **20** (wogonin 5-*O*-glucoside), and **38** (astragalin, kaempferol 3-*O*-glucoside) in GuApiGT (Supplementary Figure 141).

The active site of GuApiGT shapes like a “hammer”. Compound **2** can fit this shape very well, but its aglycone isoliquiritigenin (**2'**) only fits the handle region of this

“hammer”. Thus, **2'** is not stable in the active pocket. We further conducted 100-ns MD simulations of GuApiGT/UDP-Api/**2** and GuApiGT/UDP-Api/**2'**. The MM/GBSA binding free energy of **2'** (-58.7 ± 5.3 kcal/mol) in the active pocket is significant higher than **2** (-78.8 ± 4.2 kcal/mol). These data explained why GuApiGT could not accept free aglycones as sugar acceptor. Compound **20** (5-*O*-glycoside) shares a similar binding mode with compound **2**, while compound **38** (3-*O*-glycoside) exhibits great steric hindrance in fitting the handle region. The above descriptions have been added in Supplementary Figure 141.

Supplementary Fig. 141 Binding modes of compounds **2** (thick green sticks), **2'** (thick yellow sticks), **20** (thick magenta sticks) and **38** (thick cyan sticks) in GuApiGT. UDP-Api and important residues are depicted as thick pink sticks and grey thin sticks, respectively. The shape of the binding pocket is depicted by surface with the transparency of 0.4. **a**, top view; **b**, front view.

Given the very limited number of apiosyltransferases that have been reported, it is hard to intensively interpret their catalytic mechanisms. Recently, we have discovered a new ApiGT, which could catalyze the apiosylation of flavone 3-*O*-glycosides (Fig. R12). GuApiGT does not have this function. This new enzyme will be published elsewhere in the future. We believe more mechanisms will be interpreted as more ApiGTs are discovered.

Fig. R12 Functional characterization and substrate promiscuity of a new ApiGT which can catalyze flavone 3-*O*-glycosides.

5. While obtaining crystal structures is commendable, a comparison of GuApiGT with other GTs to understand why they cannot perform apiosyltransferase functions would be very insightful.

R: We have analyzed all the crystal structures of reported GTs. We found that a part of UDP-sugar binding region of GuApiGT is remarkably different from other GTs. This region is composed of the R³⁶⁸L³⁶⁹G³⁷⁰S³⁷¹ loop and the start of α helix (D³⁷²H³⁷³), which form a large secondary structure compared with other GTs (Fig. 4d). Our study also proved the significance of this motif for sugar donor selectivity by structural analysis and site-directed mutagenesis.

Fig. 4d The sugar binding region in crystal structures of representative ApiGT, GlcGT (GgCGT, PDB ID: 6L5P), AraGT (SbCGTb, PDB ID: 6LFZ), and RhaGT (UGT89C1, PDB ID: 6IJA). The RLGSDH motif in GuApiGT is highlighted as yellow sticks.

6. Performing enzymatic activity assays with the mutant enzyme in the presence of UDP-Xyl and UDP-Glc would offer direct evidence of substrate turnover, greatly reinforcing your claim of substrate binding.

R: Thank you for this constructive suggestion. We selected two double mutants and two triple mutants, which showed preference for UDP-Xyl and UDP-Glc, respectively. These mutant enzymes were co-incubated with isoliquiritin (**2**), UDP-Xyl, and UDP-Glc. At the presence of two sugar donors, L369/H373Q and S371/H373Q mutants showed obvious preference for UDP-Xyl, whereas I136T/L369/H373Q and I136T/S371/H373Q mutants showed obvious preference for UDP-Glc. These results further confirmed the sugar donor selectivity of mutant enzymes. These data have been added in **Supplementary Fig. 96**.

Supplementary Fig. 96 Sugar donor preference of GuApiGT mutants. **a**, conversion rates of four representative mutants. Compound **2** was used as the sugar acceptor, and both UDP-Xyl and UDP-Glc were added into the catalysis system as sugar donor. **b**, HPLC chromatograms of the enzyme catalysis products. For compounds identification of **2b** and **2c**, please see **Supplementary Fig. 93** and **Supplementary Fig. 95**.

7. The significant $\Delta\Delta E$ for GLU272 warrants its inclusion in mutagenesis studies. Also, the significance of the bold font in Supplementary Table 7 should be clarified. Lastly,

consistency in amino acid notation between the main text and supplementary information is necessary.

R: E272 is another important amino acid residue involved in the apiosylation reaction catalyzed by GuApiGT, according to QM/MM calculations. E272 could form a stable hydrogen bond with the UDP part of UDP-Api (Supplementary Fig. 91a). When we mutated E272 into alanine, the obtained E272A mutant lost its catalytic activity (Supplementary Fig. 91b). Thus, E272 plays an important role in the glycosylation process, though it is not a key residue in recognizing UDP-Api. Following the reviewer's suggestion, we have clarified the significance of the bold font as well as the consistency of amino acid notation in Supplementary Table 7: "The important residues that has an impact of more 10 kcal/mol on the original activation energy are highlighted in bold font."

Supplementary Fig. 91 a, The binding conformation of UDP-Api during apiosylation in GuApiGT. The hydrogen-bond interactions are shown as yellow dashes. **b**, HPLC analysis of E272A catalyzed product using **1** as the substrate. UDP-Api was produced by adding UDP-GlcA, purified UAXS, and NAD⁺ to the mixed system. The chromatographic peak with a retention time of around 2 min is the protein peak.

Additional Minor Revisions:

1. This sentence needs revision: "In this work, we report the first apiosyltransferase GuApiGT, which from medicinal plant *Glycyrrhiza uralensis*." A potential revision

could be: "In this work, we report the discovery of GuApiGT, the first apiosyltransferase derived from the medicinal plant *Glycyrrhiza uralensis*."

R: We have revised this sentence into "We identified the first phenolic apiosyltransferase GuApiGT from *Glycyrrhiza uralensis*". While UGT73CY2 was recently published as a triterpenoid apiosyltransferase, GuApiGT should be the first apiosyltransferase that catalyzes flavonoids and other phenolic substrates.

2. Several lines seem to lack necessary citations. Please add the appropriate references to lines 42, 50, 52, 71, and 370.

R: We have added reference 2 "*Ann Chim Phys* 1843, 9, 250-252." to line 42. Reference 1 has been added to lines 50-52. Line 71 has cited References 14. References 43 and 44 have been added to line 389 (line 370, previous version).

3. In lines 57-62, the identification of *Glycyrrhiza uralensis* Fisch. as the target plant could be more solidly substantiated, considering the existence of multiple popular medicinal plants within the Leguminosae family. Also, the discussion about liquiritin apioside could be made more relatable to lines 84-86.

R: The fresh plant was collected by the authors from Inner Mongolia Autonomous Region of China, which is the traditional cultivation region for *G. uralensis*. The plant species was identified as *Glycyrrhiza uralensis* Fisch. but not closely related species, according to the ITS2 sequence by a DNA barcoding method established by our group (Song W, *et al. Anal. Chem.* 2017, 89, 3146-3153) (Fig. R13).

G. uralensis was studied in this work because of the high amount of liquiritin apioside (>0.5% of the dry weight). The high yield strongly suggested the presence of apiosyltransferases in this plant. The related discussion was added to line 85-86 (lines 84-86, previous version).

Fig. R13 ITS2 sequence alignment of *G. uralensis*, *G. inflata*, and *G. glabra*. The key bases 159 and 383-385 are marked in blue. This figure was produced with ENDscript (<http://multalin.toulouse.inra.fr/multalin/>).

4. In Figure 4d, the electron density map appears convoluted, which may confuse the readers. Consider refining it for enhanced clarity.

R: We have removed the electron density map and have improved Fig 4d.

Fig. 4d The sugar binding region in crystal structures of representative ApiGT, GlcGT (GgCGT, PDB ID: 6L5P), AraGT (SbCGTb, PDB ID: 6LFZ), and RhaGT (UGT89C1, PDB ID: 6IJA). The RLGS DH motif in GuApiGT is highlighted as yellow sticks.

5. In Figures 4e and 4g, the rationale behind presenting some residues as lines and some as sticks is not evident. Furthermore, the coloring of R368 in orange is not explained in the figure legend.

R: We have revised Figures 4e and 4g, and have added detailed information in the legend.

Fig. 4e A representative configuration of GuApiGT/UDP-Api/2 extracted from MD simulations. The hydrogen-bond interactions and π - π /cation- π interactions are shown as yellow and purple dashes, respectively. The key amino acids interacted with ligands are highlighted using sticks. The unique R368 is depicted as orange sticks, the others as blue. The other amino acids in key motif are depicted using lines.

Fig. 4g QM/MM optimized geometry of transition state (TS) at the theory of B3LYP/6-311++g(2d,2p): amber with the electronic embedding scheme and thermal zero-point energy calculated from the theory of B3LYP/6-31g(d): amber. The QM region atoms, hydrogen bonds, and key angle and distances are highlighted in green sticks, yellow dashes, and magenta dashes, respectively. The MM region atoms are depicted using lines.

6. The term "directed evolution" might not accurately describe your engineering efforts. Perhaps "rational design" or "protein engineering" would be more fitting.

R: We have revised it into "protein engineering".

7. A clearer statement of the structural superimposition could be "The R368L369G370S371D372H373 motif in GuApiGT is mapped to F372F373G374D375Q376 of Sb3GT1."

R: For easy understanding, we have revised this sentence into "The RLGSDH (368-373) motif in GuApiGT is mapped to FFGDQ (372-376) of Sb3GT1."

8. The error bar in Figure 5c appears inconsistent with others. Please review and rectify if necessary.

R: We have revised error bars in all the figures according to the journal instructions.

9. The notation "375S" could potentially confuse the readers. I recommend providing further clarification in line 327, 330, Figure 6c, and Supplementary Figure 148.

R: To avoid any confusion, we have added this description “Based on structural analysis, we inserted a serine residue into the motif and constructed the 375S/Q377H mutant of Sb3GT1, as well as the F372R/Q376H and F372R/375S/Q377H mutants.”.

10. The differences in visualization methods used in Fig. 6a and 6c (sticks vs. lines) and the presence of a white cartoon in the zoomed image of Fig. 6a should be adequately justified in the figure legends.

R: Both the sticks in Fig.6a and the lines in Fig.6c represent amino acids. The reason for using lines is due to clearly show the key motif and the comparison between native and 375S/Q377H. We have improved the figure legends for Figures 6a and 6c according to the reviewer’s suggestion.

Fig. 6 a, The crystal structure of Sb3GT1 (PDB ID: 8IOE) and the FFGDQ (372-376) motif. The image on the right is an enlargement of the dashed rectangle, where the red part represents CTD and the grey part represents NTD. The amino acids in key motif of Sb3GT1 are highlighted using sticks. **c**, The crystal structure of Sb3GT1 375S/Q377H mutant (PDB ID: 8IOD) and superimposition of its key motif to that in wild type. The amino acids in key motif of 375S/Q377H mutant are depicted using lines.

11. The quench procedure is described differently in line 520 ("ice-cold methanol") and line 545 ("pre-cooled methanol (MeOH)"). Please ensure consistency. Additionally, providing an abbreviation for methanol (MeOH) in line 545 seems redundant, as it was not used subsequently.

R: We used pre-cooled methanol throughout the manuscript. We have deleted the abbreviation (MeOH).

12. In line 552, the statement would be clearer as "two-fold volume of methanol."

R: We have revised this sentence according to the suggestion.

13. For Extended Data Figure 1, specifying the contour level and carve radius used for the electron density maps would greatly improve clarity and interpretability. This information would aid the readers in assessing the quality of the depicted substrate-protein interactions.

R: The mesh maps did not mean electron density maps. It means the active pocket for sugar part of UDP-sugar. We have added related descriptions to the figure legend (Supplementary Fig. 138).

14. In Supplementary Fig.2, at least three impurities are present in lane 1, which contradicts the claim of achieving 95% purity.

R: We have deleted this sentence, though lane 1 showed acceptable protein purity for functional characterization.

15. Despite the ambiguity surrounding the effect of Ba²⁺ and Cu²⁺ on GuApiGT activity, it is inaccurate to describe the enzyme as divalent ion-independent. Please also ensure charges are correctly superscripted in figures.

R: We have revised this sentence into “Some divalent cations could suppress the catalytic activities (Supplementary Fig. 4d).” Moreover, we have revised the axis labels of Supplementary Fig. 4d.

Supplementary Fig. 4d Effects of divalent metal ions on enzyme activity of GuApiGT. Some divalent cations could suppress the catalytic activities.

16. Clarification of the peak proximal to 2a in Supplementary Fig. 137 is necessary. Is it a product of GuApiGT or something else?

R: The peak proximal to 2a is an unknown interfering peak. We repeated this experiment, and the reaction mixture showed only one product peak 2a. Thus, we have updated Supplementary Fig. 144.

Supplementary Fig. 144 Extracted ion chromatograms (XICs) exhibiting the presence and absence of 2a in the GuApiGT group and no GuApiGT group.

REVIEWERS' COMMENTS

Reviewer #1 (Remarks to the Author):

I believe the authors have responded well to the reviewers comments. I favor publication in Nature Communications.

Reviewer #2 (Remarks to the Author):

The Authors have addressed all of my concerns with the original manuscript.

Reviewer #3 (Remarks to the Author):

This revised manuscript has undergone impressive revisions guided by the provided feedback, bringing it notably closer to the publication standards of Nature Communications. However, there is one alteration in Supplementary Fig. 5 that should be addressed. Despite the authors' mention of conducting the experiment in triplicates, each substrate concentration point on the graph appears singular. To enhance clarity and precision, it is customary to depict each substrate concentration with either three separate points (for each replicate) or a solitary point accompanied by error bars (illustrating standard deviation or standard error of the mean).

Responses to Reviewers' comments

Reviewer #1:

I believe the authors have responded well to the reviewers comments. I favor publication in *Nature Communications*.

R: We thank the reviewer for carefully reading our manuscript, and giving us the valuable suggestions and comments to improve this manuscript.

Reviewer #2:

The Authors have addressed all of my concerns with the original manuscript.

R: We thank the reviewer for carefully reading our manuscript, and giving us the valuable suggestions and comments to improve this manuscript.

Reviewer #3:

This revised manuscript has undergone impressive revisions guided by the provided feedback, bringing it notably closer to the publication standards of *Nature Communications*. However, there is one alteration in Supplementary Fig. 5 that should be addressed. Despite the authors' mention of conducting the experiment in triplicates, each substrate concentration point on the graph appears singular. To enhance clarity and precision, it is customary to depict each substrate concentration with either three separate points (for each replicate) or a solitary point accompanied by error bars (illustrating standard deviation or standard error of the mean).

R: We thank the reviewer for carefully reading our manuscript, and giving us the valuable suggestions and comments to improve this manuscript. All the data points shown in **Supplementary Fig. 5** represent the mean of three biologically independent samples ($n=3$), and the error bars are shown as SEM (standard error of the mean). Because the SEM values were very small, the error bars could not be clearly seen in the graph. During this revision, we have replotted the figure by using the SD values (standard deviation) to illustrate the error bars. Please see the **new Supplementary Fig. 5**. The raw data are given in **Table R1**, and are also provided in a Source Data file.

New Supplementary Fig. 5 Determination of kinetic parameters for recombinant GuApiGT. The apparent K_m value was calculated from Michaelis-Menten plot with varying concentrations of compound **2** (isoliquiritin). Data are presented as mean values \pm SD ($n=3$ biologically independent samples). The source data are provided in a Source Data file.

Previous version of Supplementary Fig. 5 Determination of kinetic parameters for recombinant GuApiGT ($n=3$). The apparent K_m value was calculated from Michaelis-Menten plot with varying concentrations of compound **2** (isoliquiritin). Data are presented as mean values \pm SEM ($n=3$ biologically independent samples). The source data underlying figures are provided in a Source Data file.

Table R1. Raw data for the determination of kinetic parameters

Concentration ($\mu\text{mol}\cdot\text{L}^{-1}$)	V ($\mu\text{mol}\cdot\text{min}^{-1}\cdot\text{mg}^{-1}$)		
	replicate 1	replicate 2	replicate 3
1	0.0287	0.028333	0.027633
2.5	0.056583	0.053917	0.056333
5	0.091333	0.091167	0.091167
10	0.105333	0.098	0.103
30	0.128	0.118	0.122
40	0.122667	0.117333	0.118667
60	0.11	0.116	0.114
80	0.112	0.117333	0.114667
150	0.12	0.12	0.125